# Dynamic Causal Bayesian Optimization

**Virginia Aglietti**[*]
University of Warwick
The Alan Turing Institute
V.Aglietti@warwick.ac.uk

**Neil Dhir**[*]
The Alan Turing Institute
ndhir@turing.ac.uk

**Javier González**
Microsoft Research Cambridge
Gonzalez.Javier@microsoft.com

**Theodoros Damoulas**
University of Warwick
The Alan Turing Institute
T.Damoulas@warwick.ac.uk

## Abstract

This paper studies the problem of performing a sequence of optimal interventions in a causal dynamical system where both the target variable of interest and the inputs evolve over time. This problem arises in a variety of domains e.g. system biology and operational research. Dynamic Causal Bayesian Optimization (DCBO) brings together ideas from sequential decision making, causal inference and Gaussian process (GP) emulation. DCBO is useful in scenarios where all causal effects in a graph are changing over time. At every time step DCBO identifies a local optimal intervention by integrating both observational and past interventional data collected from the system. We give theoretical results detailing how one can transfer interventional information across time steps and define a dynamic causal GP model which can be used to quantify uncertainty and find optimal interventions in practice. We demonstrate how DCBO identifies optimal interventions faster than competing approaches in multiple settings and applications.

## 1 Introduction

Solving decision making problems in a variety of domains requires understanding of cause-effect relationships in a system. This can be obtained by experimentation. However, deciding how to intervene at every point in time is particularly complex in dynamical systems, due to the evolving nature of causal effects. For instance, companies need to decide *how* to allocate scarce resources across different quarters. In system biology, scientists need to identify genes to knockout at specific points in time. This paper describes a probabilistic framework that finds such optimal interventions over time.

Focusing on a specific example, consider a setting in which $Y_t$ denotes the unemployment-rate of an economy at time $t$, $Z_t$ is the economic growth and $X_t$ the inflation rate. Fig. 1a depicts the causal graph [26] representing an agent's understanding of the causal links between these variables. The agent aims to determine, at each time step $t \in \{0, 1, 2\}$, the optimal action to perform in order to minimize the *current* unemployment rate

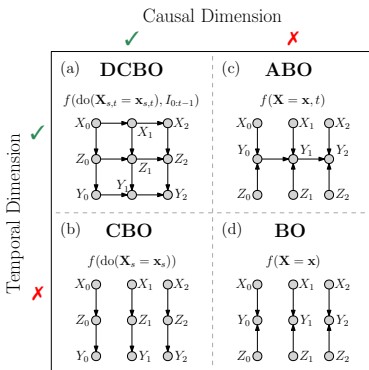

Causal Dimension

Figure 1: DAG representation of a dynamic causal global optimisation (DCGO) problem (a) and the DAG considered when using CBO, ABO or BO to address the same problem. Shaded nodes gives observed variables while the arrows represent causal effects.

---

[*]Denotes equal contribution

35th Conference on Neural Information Processing Systems (NeurIPS 2021).

$Y_t$ while accounting for the intervention cost. The investigator could frame this setting as a sequence of global optimization problems and find the solutions by resorting to Causal Bayesian Optimization [CBO, 2]. CBO extends Bayesian Optimization [BO, 30] to cases in which the variable to optimize is part of a causal model where a sequence of interventions can be performed. However, CBO does not account for the system's temporal evolution thus breaking the time dependency structure existing among variables (Fig. 1b). This will lead to sub-optimal solutions, especially in non-stationary scenarios. The same would happen when using Adaptive Bayesian Optimization [ABO, 25] (Fig. 1c) or BO (Fig. 1d). ABO captures the time dependency of the objective function but neither considers the causal structure among inputs nor their temporal evolution. BO disregards both the temporal and the causal structure. Our setting differs from both reinforcement learning (RL) and the multi-armed bandits setting (MAB). Differently from MAB we consider interventions on continuous variables where the dynamic target variable has a non-stationary interventional distribution. In addition, compared to RL, we do not model the state dynamics explicitly and allow the agent to perform a number of explorative interventions which do not change the underlying state of the system, before selecting the optimal action. We discuss these points further in §1.1.

Dynamic Causal Bayesian Optimization[2], henceforth DCBO, accounts for both the causal relationships among input variables and the causality between inputs and outputs which might evolve over time. DCBO integrates CBO with dynamic Bayesian networks (DBN), offering a novel approach for decision making under uncertainty within dynamical systems. DBN [19] are commonly used in time-series modelling and carry dependence assumptions that do not imply causation. Instead, in probabilistic causal models [27], which form the basis for the CBO framework, graphs are buildt around causal information and allow us to reason about the effects of different interventions. By combining CBO with DBNs, the proposed methodology finds an optimal *sequence* of interventions which accounts for the causal temporal dynamics of the system. In addition, DCBO takes into account past optimal interventions and transfers this information across time, thus identifying the optimal intervention faster than competing approaches and at a lower cost. We make the following contributions:

- We formulate a new class of optimization problems called Dynamic Causal Global Optimization (DCGO) where the objective functions account for the temporal causal dynamics among variables.
- We give theoretical results demonstrating how interventional information can be transferred across time-steps depending on the topology of the causal graph.
- Exploiting our theoretical results, we solve the optimization problem with DCBO. At every time step, DCBO constructs surrogate models for different intervention sets by integrating various sources of data while accounting for past interventions.
- We analyze DCBO performance in a variety of settings comparing against CBO, ABO and BO.

## 1.1 Related Work

**Dynamic Optimization** Optimization in dynamic environments has been studied in the context of evolutionary algorithms [14, 16]. More recently, other optimization techniques [28, 32, 10] have been adapted to dynamic settings, see e.g. [9] for a review. Focusing on BO, the literature on dynamic settings [3, 7, 25] is limited. The dynamic BO framework closest to this work is given by Nyikosa et al. [25] and focuses on functions defined on continuous spaces that follow a more complex behaviour than a simple Markov model. ABO treats the inputs as fixed and not as random variables, thereby disregarding their temporal evolution and, more importantly, breaking their causal dependencies.

**Causal Optimization** Causal BO [CBO, 2] focuses instead on the causal aspect of optimization and solves the problem of finding an optimal intervention in a DAG by modelling the intervention functions with single GPs or a multi-task GP model [1]. CBO disregards the existence of a temporal evolution in both the inputs and the output variable, treating them as i.i.d. overtime. While disregarding time significantly simplifies the problem, it prevents the identification of an optimal intervention at every $t$.

**Bandits and RL** In the broader decision-making literature, causal relationships have been considered in the context of bandits [4, 20–22] and reinforcement learning [23, 8, 13, 36, 24]. In these cases, actions or arms, correspond to interventions on a causal graph where there exists complex relationships between the agent's decisions and the received rewards. While dynamic settings have been considered in acausal bandit algorithms [5, 33, 35], causal MAB have focused on static settings. Dynamic settings are instead considered by RL algorithms and formalized through Markov decision processes (MDP).

---

[2]A Python implementation is available at: `https://github.com/neildhir/DCBO`.

In the current formulation, DCBO does not consider an MDP as we do not have a notion of *state* and therefore do not require an explicit model of its dynamics. The system is fully specified by the causal model. As in BO, we focus on identifying a set of time-indexed optimal actions rather than an optimal policy. We allow the agent to perform explorative interventions that do not lead to state transitions. More importantly, differently from both MAB and RL, *we allow for the integration of both observational and interventional data*. An expanded discussion on the reason why DCBO should be used and the links between DCBO, CBO, ABO and RL is included in the supplement (§8).

## 2 Background and Problem Statement

Let random variables and values be denoted by upper-case and lower-case letters respectively. Vectors are represented shown in bold. $\mathrm{do}(X = x)$ represents an intervention on $X$ whose value is set to $x$. $p(Y \mid X = x)$ represents an observational distribution and $p(Y \mid \mathrm{do}(X = x))$ represents an interventional distribution. This is the distribution of $Y$ obtained by intervening on $X$ and fixing its value to $x$ in the data generating mechanism (see Fig. 2), irrespective of the values of its parents. Evaluating $p(Y \mid \mathrm{do}(X = x))$ requires "real" interventions while $p(Y \mid X = x)$ only requires "observing" the system. $\mathcal{D}^O$ and $\mathcal{D}^I$ denote observational and interventional datasets respectively. Consider a structural causal model (SCM) defined in Definition 1.

**Definition 1. (Structural Causal Model)** [27, p. 203]. A structural causal model $M$ is a triple $\langle \mathbf{U}, \mathbf{V}, F) \rangle$ where $\mathbf{U}$ is a set of background variables (also called *exogenous*), that are determined by factors outside of the model. $\mathbf{V}$ is a set $\{V_1, V_2, \ldots, V_{|\mathbf{V}|}\}$ of observable variables (also called *endogenous*), that are determined by variables in the model (i.e., determined by variables in $\mathbf{U} \cup \mathbf{V}$). $F$ is a set of functions $\{f_1, f_2, \ldots, f_n\}$ such that each $f_i$ is a mapping from the respective domains of $U_i \cup \mathrm{Pa}(V_i)$ to $V_i$, where $U_i \subseteq \mathbf{U}$ and $\mathrm{Pa}(V_i) \subseteq \mathbf{V} \setminus V_i$ and the entire set $F$ forms a mapping from $\mathbf{U}$ to $\mathbf{V}$. In other words, each $\{f_i \in v_i \leftarrow f_i(\mathrm{Pa}(v_i), u_i) \mid i = 1, \ldots, n\}$, assigns a value to $V_i$ that depends on the values of the select set of variables $(U_i \cup \mathrm{Pa}(V_i))$.

$M$ is associated to a directed acyclic graph (DAG) $\mathcal{G}$, in which each node corresponds to a variable and the directed edges point from members of $\mathrm{Pa}(V_i)$ and $U_i$ to $V_i$. *We assume $\mathcal{G}$ to be known* and leave the integration with causal discovery [15] methods for future work. Within $\mathbf{V}$, we distinguish between three different types of variables: non-manipulative variables $\mathbf{C}$, which cannot be modified, treatment variables $\mathbf{X}$ that can be set to specific values and output variable $Y$ that represents the agent's outcome of interest. Exploiting the rules of do-calculus [27] one can compute $p(Y \mid \mathrm{do}(X = x))$ using observational data. This often involves evaluating intractable integrals which can be approximated by using observational data to get a Monte Carlo estimate $\widehat{p}(Y \mid \mathrm{do}(X = x)) \approx p(Y \mid \mathrm{do}(X = x))$. These approximations will be consistent when the number of samples drawn from $p(\mathbf{V})$ is large.

**Causality in time** One can encode the existence of causal mechanisms across time steps by explicitly representing these relationships with edges in an extended graph denoted by $\mathcal{G}_{0:T}$. For instance, the DAG in Fig. 1(a) can be seen as one of the DAGs in Fig. 1(b) propagated in time. The DAG in Fig. 1(a) captures both the causal structure existing across time steps and the causal mechanism within every "time-slice" $t$ [19]. In order to reason about interventions that are implemented in a sequential manner, that is *at time $t$ we decide which intervention to perform in the system* and so define:

**Definition 2.** $M_t$ is the SCM at time step $t$ defined as $M_t = \langle \mathbf{U}_{0:t}, \mathbf{V}_{0:t}, \mathbf{F}_{0:t} \rangle$ where $0 : t$ denotes the union of the corresponding variables or functions up to time $t$ (see Fig. 2). $\mathbf{V}_{0:t}$ includes $\mathbf{X}_{0:t} = \mathbf{X}_t$, $\mathbf{Y}_{0:t} = Y_t$ and $\mathbf{C}_{0:t} = \mathbf{C}_t \cup \mathbf{C}_{0:t-1}$. The functions in $\mathbf{F}_{0:t}$ corresponding to intervened variables are replaced by constant values while the exogenous variables related to them are excluded from $\mathbf{U}_{0:t}$.

**Definition 3.** $\mathcal{G}_t$ is the causal graph associated to $M_t$. In $\mathcal{G}_t$, the incoming edges in variables intervened at $0 : t - 1$ are mutilated while intervened variables are represented by deterministic nodes (squares) – see Fig. 2.

**Dynamic Causal Global Optimization (DCGO)** The goal of this work is to find a sequence of interventions, optimizing a target variable, *at each time step*, in a causal DAG. Given $\mathcal{G}_t$ and $M_t$, at every time step $t$, we wish to optimize $Y_t$ by intervening on a subset of the manipulative variables $\mathbf{X}_t$. The optimal intervention variables $\mathbf{X}_{s,t}^{\star}$ and intervention levels $\mathbf{x}_{s,t}^{\star}$ are given by:

$$\mathbf{X}_{s,t}^{\star}, \mathbf{x}_{s,t}^{\star} = \underset{\mathbf{X}_{s,t} \in \mathcal{P}(\mathbf{X}_t), \mathbf{x}_s \in D(\mathbf{X}_{s,t})}{\arg\min} \mathbb{E}[Y_t \mid \mathrm{do}(\mathbf{X}_{s,t} = \mathbf{x}_{s,t}), \mathbb{1}_{t>0} \cdot I_{0:t-1}] \tag{1}$$

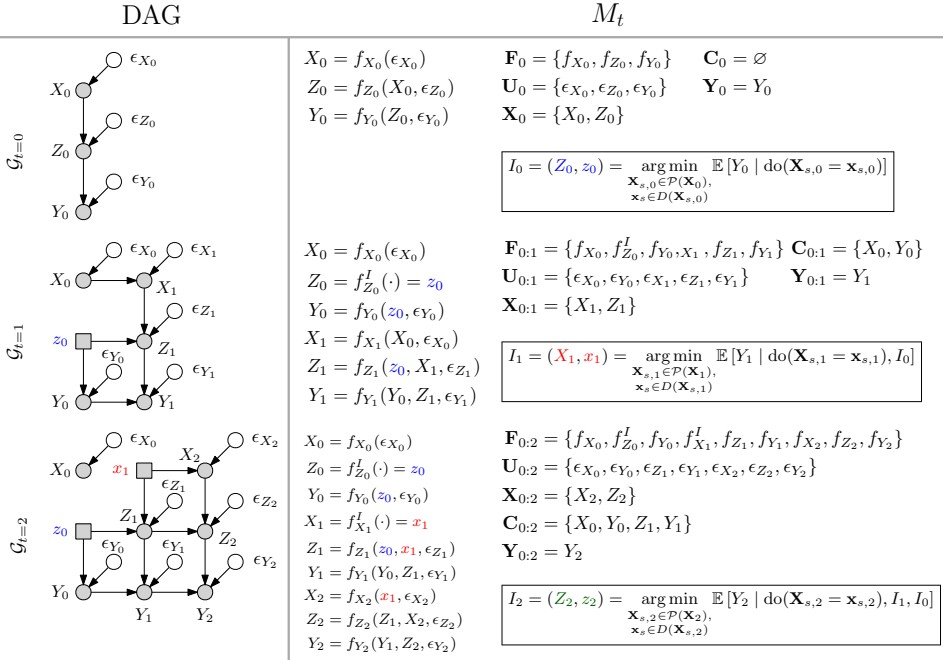

Figure 2: Structural equation models considered by DCBO at every time step $t \in \{0, 1, 2\}$. Exogenous noise variables $\epsilon_i$ are depicted here but are omitted in the remainder of the paper, to avoid clutter. For every $t$, $\mathcal{G}_t$ is a mutilated version of $\mathcal{G}_{t-1}$ reflecting the optimal intervention implemented in the system at $0 : t - 1$ which are represented by squares. The SCM functions in $\mathbf{F}_{0:t}$ corresponding to the intervened variables are set to constant values. The exogenous variables that only related to the intervened variables are excluded from $U_t$. $\mathbf{C}_{0:t}$ is given by the set $\{\mathbf{C}_t \cup \mathbf{C}_{0:t-1} \cup \mathbf{Y}_{0:t-1} \cup \mathbf{X}_{0:t-1}\}$.

where $I_{0:t-1} = \bigcup_{i=0}^{t-1} \mathrm{do}\big(\mathbf{X}_{s,i}^{\star} = \mathbf{x}_{s,i}^{\star}\big)$ denotes previous interventions, $\mathbb{1}_{t>0}$ is the indicator function and $\mathcal{P}(\mathbf{X}_t)$ is the power set of $\mathbf{X}_t$. $D(\mathbf{X}_{s,t})$ represents the interventional domain of $\mathbf{X}_{s,t}$. In the sequel we denote the previously intervened variables by $I_{0:t-1}^V = \bigcup_{i=0}^{t-1} \mathbf{X}_{s,i}^{\star}$ and implemented intervention levels by $I_{0:t-1}^L = \bigcup_{i=0}^{t-1} \mathbf{x}_{s,i}^{\star}$. The cost of each intervention is given by $\mathrm{cost}(\mathbf{X}_{s,t}, \mathbf{x}_{s,t})$. In order to solve the problem in Eq. (1) we make the following assumptions :

**Assumptions 1.** Denote by $\mathcal{G}(t)$ the causal graph including variables at time $t$ in $\mathcal{G}_{0:T}$ and let $Y_t^{\mathrm{PT}} = \mathrm{Pa}(Y_t) \cap Y_{0:t-1}$ be the set of variables in $\mathcal{G}_{0:T}$ that are both parents of $Y_t$ and targets at previous time step. Let the set $Y_t^{\mathrm{PNT}} = \mathrm{Pa}(Y_t) \setminus Y_t^{\mathrm{PT}}$ be the complement and denote by $f_{Y_t}(\cdot)$ the functional mapping for $Y_t$ in $M_t$. We make the following assumptions:

1. Invariance of causal structure: $\mathcal{G}(t) = \mathcal{G}(0), \forall t > 0$.

2. Additivity of $f_{Y_t}(\cdot)$ that is $Y_t = f_{Y_t}(\mathrm{Pa}(Y_t)) + \epsilon$ with $f_{Y_t}(\mathrm{Pa}(Y_t)) = f_Y^Y(Y_t^{\mathrm{PT}}) + f_Y^{\mathrm{NY}}(Y_t^{\mathrm{PNT}})$ where $f_Y^Y$ and $f_Y^{\mathrm{NY}}$ are two generic unknown functions and $\epsilon \sim \mathcal{N}(0, \sigma^2)$.

3. Absence of unobserved confounders in $\mathcal{G}_{0:T}$.

Assumption (3) implies the absence of unobserved confounders at every time step. For instance, this is the case in Fig. 1a. Still in this DAG, Assumption (2) implies $f_{Y_t}(\mathrm{Pa}(Y_t)) = f_Y^Y(Y_{t-1}) + f_Y^{\mathrm{NY}}(Z_t) + \epsilon_{Y_t}, \forall t > 0$. Finally, Assumption (1) implies the existence of the same variables at every time step and a constant orientation of the edges among them for $t > 0$.

Notice that Assumptions 1 imply invariance of the causal structure *within* each time-slice, i.e. the structure, edges and vertices, concerning the nodes with the same time index. This means that, across time steps, both the graph and the functional relationships can change. Therefore, not only can the causal effects change significantly across time steps, but also the input dimensionality of the causal functions we model, might change. For instance, in the DAG of Fig. 3(c), the target function for $Y_2$ has dimensionality 3 and a function $f_{Y_t}(\cdot)$ that is completely different from the one assumed for $Y_1$ that has only two parents. We can thus model a wide variety of settings and causal effects despite this

assumption. Furthermore, even though we assume an additive structure for the functional relationship on $Y$, the use of GPs allow us to have flexible models with highly non-linear causal effects across different graph structures. In the causality literature, GP models are well established and have shown good performances compared to parametric linear and non-linear models (see e.g. [31, 34, 37]). The sum of GPs gives a flexible and computationally tractable model that can be used to capture highly non-linear causal effects while helping with interpretability [12, 11].

# 3 Methodology

In this section we introduce Dynamic Causal Bayesian Optimization (DCBO), a novel methodology addressing the problem in Eq. (1). We first study the correlation among objective functions for two consecutive time steps and use it to derive a recursion formula that, based on the topology of the graph, expresses the causal effects at time $t$ as a function of previously implemented interventions (see square nodes in Fig. 2). Exploiting these results, we develop a new surrogate model for the objective functions that can be used within a CBO framework to find the optimal sequence of interventions. This model enables the integration of observational data and interventional data, collected at previous time-steps and interventional data collected at time $t$, thereby accelerating the identification of the current optimal intervention.

## 3.1 Characterization of the time structure in a DAG with time dependent variables

The following result provides a theoretical foundation for the dynamic causal GP model introduced later. In particular, it derives a recursion formula allowing us to express the objective function at time $t$ as a function of the objective functions corresponding to the optimal interventions at previous time steps. The proof is given in the appendix (§2).

**Definition 4.** Consider a DAG $\mathcal{G}_{0:T}$ and the objective function $\mathbb{E}[Y_t \mid \mathrm{do}(\mathbf{X}_{s,t} = \mathbf{x}_{s,t}), I_{0:t-1}]$ for a generic time step $t \in \{0, \ldots, T\}$. Denote by $Y_t^{\mathrm{PT}} = (\mathrm{Pa}(Y_t) \cap Y_{0:t-1})$ the parents of $Y_t$ that are targets at previous time steps and by $Y_t^{\mathrm{PNT}} = \mathrm{Pa}(Y_t) \backslash Y_t^{\mathrm{PT}}$ the remaining parents. For any $\mathbf{X}_{s,t} \in \mathcal{P}(\mathbf{X}_t)$ and $I_{0:t-1}^V \subseteq \mathbf{X}_{0:t-1}$ we define the following sets:

- $\mathbf{X}_{s,t}^{\mathrm{PY}} = \mathbf{X}_{s,t} \cap \mathrm{Pa}(Y_t)$ includes the variables in $\mathbf{X}_{s,t}$ that are parents of $Y_t$.

- $I_{0:t-1}^{\mathrm{PY}} = I_{0:t-1}^V \cap \mathrm{Pa}(Y_t)$ includes the variables in $I_{0:t-1}^V$ that are parents of $Y_t$.

- $W \subset \mathrm{Pa}(Y_t)$ such that $\mathrm{Pa}(Y_t) = (\mathrm{Pa}(Y_t) \cap Y_{0:t-1}) \cup \mathbf{X}_{s,t}^{\mathrm{PY}} \cup I_{0:t-1}^{\mathrm{PY}} \cup W$. $W$ includes variables that are parents of $Y_t$ but are not targets nor intervened variables.

The values of $I_{0:t-1}$, $\mathbf{X}_{s,t}^{\mathrm{PY}}$, $I_{0:t-1}^{\mathrm{PY}}$ and $W$ will be denoted by $\mathbf{i}$, $\mathbf{x}^{\mathrm{PY}}$, $\mathbf{i}^{\mathrm{PY}}$ and $\mathbf{w}$ respectively.

**Theorem 1. Time operator.** Consider a DAG $\mathcal{G}_{0:T}$ and the related SCM satisfying Assumptions 1. It is possible to prove that, $\forall \mathbf{X}_{s,t} \in \mathcal{P}(\mathbf{X}_t)$, the intervention function $f_{s,t}(\mathbf{x}) = \mathbb{E}[Y_t \mid \mathrm{do}(\mathbf{X}_{s,t} = \mathbf{x}), \mathbb{1}_{t>0} \cdot I_{0:t-1}]$ with $f_{s,t}(\mathbf{x}) : D(\mathbf{X}_{s,t}) \to \mathbb{R}$ can be written as:

$$f_{s,t}(\mathbf{x}) = f_Y^Y(\mathbf{f}^\star) + \mathbb{E}_{p(\mathbf{w} \mid \mathrm{do}(\mathbf{X}_{s,t} = \mathbf{x}), \mathbf{i})}\left[f_Y^{\mathrm{NY}}(\mathbf{x}^{\mathrm{PY}}, \mathbf{i}^{\mathrm{PY}}, \mathbf{w})\right] \qquad (2)$$

where $\mathbf{f}^\star = \{\mathbb{E}[Y_i \mid \mathrm{do}(\mathbf{X}_{s,i}^\star = \mathbf{x}_{s,i}^\star), I_{0:i-1}]\}_{Y_i \in Y_t^{\mathrm{PT}}}$ that is the set of previously observed optimal targets that are parents of $Y_t$. $f_Y^Y$ denotes the function mapping $Y_t^{\mathrm{PT}}$ to $Y_t$ and $f_Y^{\mathrm{NY}}$ represents the function mapping $Y_t^{\mathrm{PNT}}$ to $Y_t$.

Eq. (2) reduces to $\mathbb{E}_{p(\mathbf{w} \mid \mathrm{do}(\mathbf{X}_{s,t} = \mathbf{x}), \mathbf{i})}\left[f_Y^{\mathrm{NY}}(\mathbf{x}^{\mathrm{PY}}, \mathbf{i}^{\mathrm{PY}}, \mathbf{w})\right]$ when $Y_t$ does not depend on previous targets. This is the setting considered in CBO that can be thus seen as a particular instance of DCBO. Exploiting Assumptions (1), it is possible to further expand the second term in Eq. (2) to get the following expression. A proof is given in the supplement (§2).

**Corollary 1.** Given Assumptions 1, we can write:

$$\mathbb{E}_{p(\mathbf{w} \mid \mathrm{do}(\mathbf{X}_{s,t} = \mathbf{x}), \mathbf{i})}\left[f_Y^{\mathrm{NY}}(\mathbf{x}^{\mathrm{PY}}, \mathbf{i}^{\mathrm{PY}}, \mathbf{w})\right] = \mathbb{E}_{p(\mathbf{U}_{0:t})}\left[f_Y^{\mathrm{NY}}(\mathbf{x}^{\mathrm{PY}}, \mathbf{i}^{\mathrm{PY}}, \{C(W)\}_{W \in \mathbf{w}})\right] \qquad (3)$$

where $p(\mathbf{U}_{0:t})$ is the distribution for the exogenous variables up to time $t$ and $C(W)$ is given by:

$$C(W) = \begin{cases} f_W(\mathbf{u}_W, \mathbf{x}^{\mathrm{PW}}, \mathbf{i}^{\mathrm{PW}}) & \text{if} \quad R = \varnothing \\ f_W(\mathbf{u}_W, \mathbf{x}^{\mathrm{PW}}, \mathbf{i}^{\mathrm{PW}}, r) & \text{if} \quad R \subseteq \mathbf{X}_{s,t} \cup I_{0:t-1}^V \\ f_W(\mathbf{u}_W, \mathbf{x}^{\mathrm{PW}}, \mathbf{i}^{\mathrm{PW}}, C(R)) & \text{if} \quad R \not\subseteq \mathbf{X}_{s,t} \cup I_{0:t-1}^V \end{cases}$$

where $f_W$ represents the functional mapping for $W$ in the SCM and $\mathbf{u}_W$ is the set of exogenous variables with edges into $W$. $\mathbf{x}^{\text{PW}}$ and $\mathbf{i}^{\text{PW}}$ are the values corresponding to $\mathbf{X}_{s,t}^{\text{PW}}$ and $I_{0:t-1}^{\text{PW}}$ which in turn represent the subset of variables in $\mathbf{X}_{s,t}$ and $I_{0:t-1}^V$ that are parents of $W$. Finally $r$ is the value of $R = \text{Pa}(W) \setminus (\mathbf{X}_{s,t}^{\text{PY}} \cup I_{0:t-1}^{\text{PW}})$.

**Examples for Eq. (2):** For the DAG in Fig. 1(a), at time $t = 1$ and with $I_{0:t-1}^V = \{Z_0\}$, we have $\mathbb{E}[Y \mid \text{do}(Z_1 = z), I_0] = f_Y^Y(y_0^\star) + f_Y^{\text{NY}}(z)$. Indeed in this case $\mathbf{W} = \varnothing$, $\mathbf{x}^{\text{PY}} = z$ and $\mathbf{f}^\star = \{y_0^\star = \mathbb{E}[Y_0 \mid \text{do}(Z_0 = z_0)]\}$. Still at $t = 1$ and with $I_{0:t-1}^V = \{Z_0\}$, the objective function for $\mathbf{X}_{s,t} = \{X_1\}$ can be written as $f_Y^Y(y_0^\star) + \mathbb{E}_{p(z_1 \mid \text{do}(X_1 = x), I_0)}\left[f_Y^{\text{NY}}(z_1)\right]$ as $\mathbf{W} = \{Z_1\}$. All derivations for these expressions and alternative graphs are given in the supplement (§2).

## 3.2 Restricting the search space

The search space for the problem in Eq. (1) grows exponentially with $|\mathbf{X}_t|$ thus slowing down the identification of the optimal intervention when $\mathcal{G}$ includes more than a few nodes. Indeed, a naive approach of finding $\mathbf{X}_{s,t}^\star$ at $t = 0, \ldots, T$ would be to explore the $2^{|\mathbf{X}_t|}$ sets in $\mathcal{P}(\mathbf{X}_t)$ at every $t$ and keep $2^{|\mathbf{X}_t|}$ models for the objective functions. In the static setting, CBO reduces the search space by exploiting the results in [21]. In particular, it identifies a subset of variables $\mathbb{M} \subseteq \mathcal{P}(\mathbf{X})$ worth intervening on thus reducing the size of the exploration set to $2^{|\mathbb{M}|}$.

In our dynamic setting, the objective functions change at every time step depending on the previously implemented interventions and one would need to recompute $\mathbb{M}$ at every $t$. However, it is possible to show that, given Assumptions 1, the search space remains constant over time. Denote by $\mathbb{M}_t$ the set $\mathbb{M}$ at time $t$ and let $\mathbb{M}_0$ represent the set at $t = 0$ which corresponds to $\mathbb{M}$ computed in CBO. For $t > 0$ it is possible to prove that:

**Proposition 3.1.** MIS **in time.** If Assumptions 1 are satisfied, $\mathbb{M}_t = \mathbb{M}_0$ for $t > 0$.

## 3.3 Dynamic Causal GP model

Here we introduce the Dynamic Causal GP model that is used as a surrogate model for the objective functions in Eq. (1). The prior parameters are constructed by exploiting the recursion in Eq. (2). At each time step $t$, the agent explores the sets in $\mathbb{M}_t \subseteq \mathcal{P}(\mathbf{X}_t)$ by selecting the next intervention to be the one maximizing a given acquisition function. The DCBO algorithm is shown in Algorithm 1.

**Prior Surrogate Model** At each time step $t$ and for each $\mathbf{X}_{s,t} \in \mathbb{M}_t$, we place a GP prior on the objective function $f_{s,t}(\mathbf{x}) = \mathbb{E}[Y_t \mid \text{do}(\mathbf{X}_{s,t} = \mathbf{x}), \mathbb{1}_{t>0} \cdot I_{0:t-1}]$. We construct the prior parameters exploiting the recursive expression in Eq. (2):

$$f_{s,t}(\mathbf{x}) \sim \mathcal{GP}(m_{s,t}(\mathbf{x}), k_{s,t}(\mathbf{x}, \mathbf{x}')) \text{ where}$$

$$m_{s,t}(\mathbf{x}) = \mathbb{E}\left[f_Y^Y(\mathbf{f}^\star) + \widehat{\mathbb{E}}[f_Y^{\text{NY}}(\mathbf{x}^{\text{PY}}, \mathbf{i}^{\text{PY}}, \mathbf{w})]\right]$$

$$k_{s,t}(\mathbf{x}, \mathbf{x}') = k_{\text{RBF}}(\mathbf{x}, \mathbf{x}') + \sigma_{s,t}(\mathbf{x})\sigma_{s,t}(\mathbf{x}') \text{ with}$$

$$\sigma_{s,t}(\mathbf{x}) = \sqrt{\mathbb{V}[f_Y^Y(\mathbf{f}^\star) + \hat{\mathbb{E}}\left[f_Y^{\text{NY}}(\mathbf{x}^{\text{PY}}, \mathbf{i}^{\text{PY}}, \mathbf{w})\right]}$$

and $k_{\text{RBF}}(\mathbf{x}, \mathbf{x}') := \exp(-\frac{||\mathbf{x} - \mathbf{x}'||^2}{2l^2})$ represents the radial basis function kernel [29]. We have it that

$$\hat{\mathbb{E}}\left[f_Y^{\text{NY}}(\mathbf{x}^{\text{PY}}, \mathbf{i}^{\text{PY}}, \mathbf{w})\right] = \hat{\mathbb{E}}_{p(\mathbf{w} \mid \text{do}(\mathbf{X}_{s,t} = \mathbf{x}), \mathbf{i})}\left[f_Y^{\text{NY}}(\mathbf{x}^{\text{PY}}, \mathbf{i}^{\text{PY}}, \mathbf{w})\right]$$

represents the expected value of $f_Y^{\text{NY}}(\mathbf{x}^{\text{PY}}, \mathbf{i}^{\text{PY}}, \mathbf{w})$ with respect to $p(\mathbf{w} \mid \text{do}(\mathbf{X}_{s,t} = \mathbf{x}), \mathbf{i})$ which is estimated via the do-calculus using observational data. The outer expectation in $m_{s,t}(\mathbf{x})$ and the variance in $\sigma_{s,t}(\mathbf{x})$ are computed with respect to $p(f_Y^Y, f_Y^{\text{NY}})$ which is also estimated using observational data. In this work we model $f_Y^Y$, $f_Y^{\text{NY}}$ and all functions in the SCM by independent GPs.

Both $m_{s,t}(\mathbf{x})$ and $\sigma_{s,t}(\mathbf{x})$ can be equivalently written by exploiting the equivalence in Eq. (3). In both cases, this prior construction allows the integration of three different types of data: observational data, interventional data collected at time $t$ and the optimal interventional data points collected in the past. The former is used to estimate the SCM model and $p(\mathbf{w} \mid \text{do}(\mathbf{X}_{s,t} = \mathbf{x}), \mathbf{i})$ via the rules of

do-calculus. The optimal interventional data points at $0 : t-1$ determine the shift $f_Y^Y(\mathbf{f}^\star)$ while the interventional data collected at time $t$ are used to update the prior distribution on $f_{s,t}(\mathbf{x})$. Similar prior constructions were previously considered in static settings [2, 1] where only observational and interventional data at the current time step were used. The additional shift term appears here as there exists causal dynamics in the target variables and the objective function is affected by previous decisions. Fig. 2 in the appendix shows a synthetic example in which accounting for the dynamic aspect in the prior formulation leads to a more accurate GP posterior compared to the baselines, especially when the the optimum location changes across time steps.

**Likelihood** Let $\mathcal{D}_{s,t}^I = (\mathbf{X}^I, \mathbf{Y}_{s,t}^I)$ be the set of interventional data points collected for $\mathbf{X}_{s,t}$ with $\mathbf{X}^I$ being a vector of intervention values and $\mathbf{Y}_{s,t}^I$ representing the corresponding vector of observed target values. As in standard BO we assume each $y_{s,t}$ in $\mathbf{Y}_{s,t}^I$ to be a noisy observation of the function $f_{s,t}(\mathbf{x})$ that is $y_{s,t}(\mathbf{x}) = f_{s,t}(\mathbf{x}) + \epsilon_{s,t}$ with $\epsilon_{s,t} \sim \mathcal{N}(0, \sigma^2)$ for $s \in \{1, \dots, |\mathbb{M}_t|\}$ and $t \in \{0, \dots, T\}$. In compact form, the joint likelihood function for $\mathcal{D}_{s,t}^I$ is $p(\mathbf{Y}_{s,t}^I \mid f_{s,t}, \sigma^2) = \mathcal{N}(f_{s,t}(\mathbf{X}^I), \sigma^2 \mathbf{I})$.

**Acquisition Function** Given our surrogate models at time $t$, the agent selects the interventions to implement by solving a Causal Bayesian Optimization problem [2]. The agent explores the sets in $\mathbb{M}_t$ and decides where to intervene by maximizing the Causal Expected Improvement (EI). Denote by $y_t^\star$ the optimal observed target

---

**Algorithm 1:** DCBO

**Data:** $\mathcal{D}^O$, $\{\mathcal{D}_{s,t=0}^I\}_{s \in \{0,\dots,|\mathbb{M}_0|\}}$, $\mathcal{G}_{0:T}$, $H$.
**Result:** Optimal intervention path
$\{\mathbf{X}_{s,t}^\star, \mathbf{x}_{s,t}^\star, y_t^\star\}_{t=1}^T$
**Initialise**: $\mathbb{M}$, $\mathcal{D}_0^I$ and initial optimal $\mathcal{D}_\star^I = \varnothing$.
**for** $t = 0, \dots, T$ **do**
  1. Initialise dynamic causal GP models for all
    $\mathbf{X}_{s,t} \in \mathbb{M}_t$ using $\mathcal{D}_{\star,t-1}^I$ if $t > 0$.
  2. Initialise interventional dataset
    $\{\mathcal{D}_{s,t}^I\}_{s \in \{0,\dots,|\mathbb{M}_t|\}}$
  **for** $h = 1, \dots, H$ **do**
    1. Compute $\mathrm{EI}_{s,t}(\mathbf{x})$ for each $\mathbf{X}_{s,t} \in \mathbb{M}_t$.
    2. Obtain $(s^\star, \alpha^\star)$
    3. Intervene and augment $\mathcal{D}_{s=s^\star,t}^I$
    4. Update posterior for $f_{s=s^\star,t}$
  **end**
  3. Return the optimal intervention $(\mathbf{X}_{s,t}^\star, \mathbf{x}_{s,t}^\star)$
  4. Append optimal interventional data
    $\mathcal{D}_{\star,t}^I = \mathcal{D}_{\star,t-1}^I \cup ((\mathbf{X}_{s,t}^\star, \mathbf{x}_{s,t}^\star), y_t^\star)$
**end**

---

value in $\{\mathbf{Y}_{s,t}^I\}_{s=1}^{|\mathbb{M}_t|}$ that is the optimal observed target across all intervention sets at time $t$. The Causal EI is given by

$$\mathrm{EI}_{s,t}(\mathbf{x}) = \mathbb{E}_{p(y_{s,t})}[\max(y_{s,t} - y_t^\star, 0)] / \mathrm{cost}(\mathbf{X}_{s,t}, \mathbf{x}_{s,t}).$$

Let $\alpha_1, \dots, \alpha_{|\mathbb{M}_t|}$ be solutions of the optimization of $\mathrm{EI}_{s,t}(\mathbf{x})$ for each set in $\mathbb{M}_t$ and $\alpha^\star := \max\{\alpha_1, \dots, \alpha_{|\mathbb{M}_t|}\}$. The next best intervention to explore at time $t$ is given by $s^\star = \mathrm{argmax}_{s \in \{1, \cdots, |\mathbb{M}_t|\}} \alpha_s$. Therefore, the set-value pair to intervene on is $(s^\star, \alpha^\star)$. At every $t$, the agent implement $H$ *explorative* interventions in the system which are selected by maximizing the Causal EI. Once the budget $H$ is exhausted, the agent implements what we call the *decision* intervention $I_t$, that is the optimal intervention found at the current time step, and move forward to a new optimization at $t + 1$ carrying the information in $y_{0:t-1}^\star$. The parameter $H$ determines the level of exploration of the system and acts as a budget for the CBO algorithm. Its value is determined by the agent and is generally problem specific.

**Posterior Surrogate Model** For any set $\mathbf{X}_{s,t} \in \mathbb{M}_t$, the posterior distribution $p(f_{s,t} \mid \mathcal{D}_{s,t}^I)$ can be derived analytically via standard GP updates. $p(f_{s,t} \mid \mathcal{D}_{s,t}^I)$ will also be a GP with parameters

$$m_{s,t}(\mathbf{x} \mid \mathcal{D}_{s,t}^I) = m_{s,t}(\mathbf{x}) + k_{s,t}(\mathbf{x}, \mathbf{X}^I)[k_{s,t}(\mathbf{X}^I, \mathbf{X}^I) + \sigma^2 \mathbf{I}](\mathbf{Y}_{s,t}^I - m_{s,t}(\mathbf{X}^I) \text{ and}$$

$$k_{s,t}(\mathbf{x}, \mathbf{x}' \mid \mathcal{D}_{s,t}^I) = k_{s,t}(\mathbf{x}, \mathbf{x}') - k_{s,t}(\mathbf{x}, \mathbf{X}^I)[k_{s,t}(\mathbf{X}^I, \mathbf{X}^I) + \sigma^2 \mathbf{I}]k_{s,t}(\mathbf{X}^I, \mathbf{x}').$$

## 4 Experiments

We evaluate the performance of DCBO in a variety of synthetic and real world settings with DAGs given in Fig. 3. We first run the algorithm for a stationary setting where both the graph structure and the SCM do not change over time (STAT.). We then consider a scenario characterised by increased observation noise (NOISY) for the manipulative variables and a settings where observational data are missing at some time steps (MISS.). Still assuming stationarity, we then test the algorithm in a DAG where there are multivariate interventions in $\mathbb{M}_t$ (MULTIV.). Finally, we run DCBO for a non-stationary graph where both the SCM and the DAG change over time (NONSTAT.). To conclude,

we use DCBO to optimize the unemployment rate of a closed economy (DAG in Fig. 3d, ECON.) and to find the optimal intervention in a system of ordinary differential equation modelling a real predator-prey system (DAG in Fig. 3e, ODE). We provide a discussion on the applicability of DCBO to real-world problems in §7 of the supplement together with all implementation details.

**Baselines** We compare against the algorithms in Fig. 1. Note that, by constructions, ABO and BO intervene on all manipulative variables while DCBO and CBO explore only $\mathbb{M}_t$ at every $t$. In addition, both DCBO and ABO reduce to CBO and BO at the first time step. We assume the availability of an observational dataset $\mathcal{D}^O$ and set a unit intervention cost for all variables.

**Performance metric** We run all experiments for 10 replicates and show the average convergence path at every time step. We then compute the values of a modified "gap" metric[3] across time steps and with standard errors across replicates. The metric is defined as

$$\mathrm{G}_t = \left[ \frac{y(\mathbf{x}^\star_{s,t}) - y(\mathbf{x}_{\mathrm{init}})}{y^\star - y(\mathbf{x}_{\mathrm{init}})} + \frac{H - H(\mathbf{x}^\star_{s,t})}{H} \right] \bigg/ \left( 1 + \frac{H-1}{H} \right) \tag{4}$$

where $y(\cdot)$ represents the evaluation of the objective function, $y^\star$ is the global minimum, and $\mathbf{x}_{\mathrm{init}}$ and $\mathbf{x}^\star_{s,t}$ are the first and best evaluated point, respectively. The term $\frac{H - H(\mathbf{x}^\star_{s,t})}{H}$ with $H(\mathbf{x}^\star_{s,t})$ denoting the number of explorative trials needed to reach $\mathbf{x}^\star_{s,t}$ captures the speed of the optimization. This term is equal to zero when the algorithm is not converged and equal to $(H-1)/H$ when the algorithm converges at the first trial. We have $0 \leq \mathrm{G}_t \leq 1$ with higher values denoting better performances. For each method we also show the average percentage of replicates where the optimal intervention set $\mathbf{X}^\star_{s,t}$ is identified.

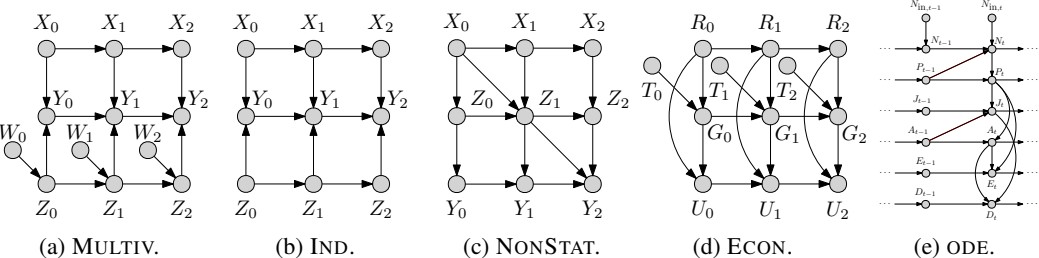

|  |  |  |  |  |
|---|---|---|---|---|
| (a) MULTIV. | (b) IND. | (c) NONSTAT. | (d) ECON. | (e) ODE. |

Figure 3: DAGs used in the experimental sections for the real (§4.2) and synthetic data (§4.1).

## 4.1 Synthetic Experiments

**Stationary DAG and SCM (STAT.)** We run the algorithms for the DAG in Fig. 1(a) with $T = 3$ and $N = 10$. *For $t > 0$,* DCBO *converges to the optimal value faster than competing approaches* (see Fig. 2 in the supplement, right panel, $3^{\mathrm{rd}}$ row). DCBO identifies the optimal intervention set in $93\%$ of the replicates (Table 2) and reaches the highest average gap metric (Table 1). In this experiment the location of the optimum changes significantly both in terms of optimal set and intervention value when going from $t = 0$ to $t = 1$. This information is incorporated by DCBO through the prior dependency on $y^\star_{0:t-1}$. In addition, ABO performance improves over time as it accumulates interventional data and uses them to fit the temporal dimension of the surrogate model. This benefits ABO in a stationary setting but might penalise it in non-stationary setting where the objective functions change significantly.

**Noisy manipulative variables (NOISY):** *The benefit of using* DCBO *becomes more apparent when the manipulative variables observations are noisy* while the evolution of the target variable is more accurately detected. In this case both the convergence of DCBO and CBO are slowed down by noisy observations which are diluting the information provided by the do-calculus making the priors less informative. However, the DCBO prior dependency on $y^\star_{0:t-1}$ allows it to correctly identify the shift in the target variable thus improving the prior accuracy and the speed-up of the algorithm (Fig. 4).

**Missing observational data (MISS.)** *Incorporating dynamic information in the surrogate model allows us to efficiently optimise a target variable even in setting where observational data are missing.*

---

[3]This metric is a modified version of the one used in [18].

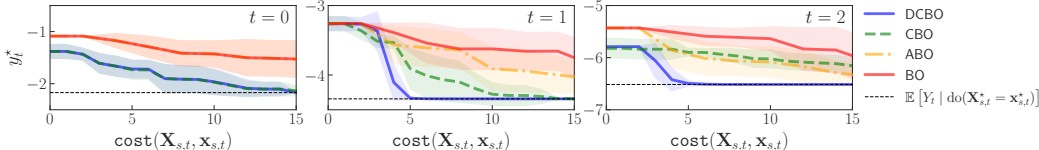

Figure 4: Experiment NOISY. Convergence of DCBO and competing methods across replicates. The dashed black line (- - -) gives the optimal outcome $y_t^*, \forall t$. Shaded areas are $\pm$ one standard deviation.

We consider the DAG in Fig. 1(a) with $T = 6$, $N = 10$ for the first three time steps and $N = 0$ afterwards. DCBO uses the observational distributions learned with data from the first three time steps to construct the prior for $t > 3$. On the contrary, CBO uses the standard prior for $t > 3$. In this setting DCBO consistently outperforms CBO at every time step. However, ABO performance improves over time and outperforms DCBO starting from $t = 4$ due to its ability to exploit all interventional data collected over time (see Fig. 3 in the supplement).

**Multivariate intervention sets (MULTIV.)** *When the optimal intervention set is multivariate, both* DCBO *and* CBO *convergence speed worsen*. For instance, for the DAG in Fig. 3a, $|\mathbb{M}| = 5$ thus both CBO and DCBO will have to perform more explorative interventions before finding the optimum. At the same time, ABO and BO consider interventions only on $\{W_t, X_t, Z_t\}, \forall t$ and need to explore an even higher intervention space. The performance of all methods decrease in this case (Table 1) but DCBO still identifies the optimal intervention set in $93\%$ of the replicates (Table 2).

**Independent manipulative variables (IND.):** *Having to explore multiple intervention sets significantly penalises* DCBO *and* CBO *when there is no causal relationship among manipulative variables* which are also the only parents of the target. This is the case for the DAG in Fig. 3b where the optimal intervention is $\{X_t, Z_t\}$ at every time step. In this case, exploring $\mathbb{M}$ and propagating uncertainty in the causal prior slows down DCBO convergence and decreases both its performance (Table 1) and capability to identify the optimal intervention set (Table 2).

**Non-stationary DAG and SCM (NONSTAT.):** DCBO *outperforms all approaches in non-stationary settings where both the* DAG *and the* SCM *change overtime* – see Fig. 3c. Indeed, DCBO can timely incorporate changes in the system via the dynamic causal prior construction while CBO, BO and ABO need to perform several interventions before accurately learning the new objective functions.

Table 1: Average $G_t$ across 10 replicates and time steps. See Fig. 1 for a summary of the baselines. Higher values are better. The best result for each experiment in bold. Standard errors in brackets.

| | Synthetic data | | | | | | Real data | |
| --- | --- | --- | --- | --- | --- | --- | --- | --- |
| | STAT. | MISS. | NOISY | MULTIV. | IND. | NONSTAT. | ECON. | ODE |
| DCBO | **0.88** | **0.84** | **0.75** | **0.49** | 0.48 | **0.69** | **0.64** | **0.67** |
| | (0.00) | (0.01) | (0.00) | (0.01) | (0.04) | (0.00) | (0.01) | (0.00) |
| CBO | 0.70 | 0.70 | 0.51 | 0.48 | 0.47 | 0.61 | 0.61 | 0.65 |
| | (0.01) | (0.02) | (0.02) | (0.09) | (0.07) | (0.00) | (0.01) | (0.00) |
| ABO | 0.56 | 0.49 | 0.49 | 0.39 | **0.54** | 0.38 | 0.57 | 0.48 |
| | (0.01) | (0.02) | (0.04) | (0.21) | (0.01) | (0.02) | (0.02) | (0.01) |
| BO | 0.54 | 0.48 | 0.38 | 0.35 | 0.50 | 0.38 | 0.50 | 0.44 |
| | (0.02) | (0.03) | (0.05) | (0.08) | (0.01) | (0.03) | (0.01) | (0.03) |

## 4.2 Real experiments

**Real-World Economic data (ECON.)** We use DCBO to minimize the unemployment rate $U_t$ of a closed economy. We consider its causal relationships with economic growth ($G_t$), inflation rate ($R_t$) and fiscal policy ($T_t$)[4]. Inspired by the economic example in [17] we consider the DAG in

---

[4]The causality between economic variables is oversimplified in this example thus the results cannot be used to guide public policy and are only meant to showcase how DCBO can be used within a real application.

Table 2: Average % of replicates across time steps for which $\mathbf{X}^{\star}_{s,t}$ is identified. See Fig. 1 for a summary of the baselines. Higher values are better. The best result for each experiment in bold.

| | Synthetic data | | | | | | Real data | |
|---|---|---|---|---|---|---|---|---|
| | STAT. | MISS. | NOISY | MULTIV. | IND. | NONSTAT. | ECON. | ODE |
| DCBO | **93.00** | 58.00 | **100.00** | **93.00** | 93.00 | **100.00** | 86.67 | **33.3** |
| CBO | 90.00 | **85.00** | 90.00 | 90.0 | 90.00 | **100.00** | **93.33** | **33.3** |
| ABO | 0.00 | 0.00 | 0.00 | 0.00 | **100.00** | 0.00 | 66.67 | 0.00 |
| BO | 0.00 | 0.00 | 0.00 | 0.00 | **100.00** | 0.00 | 66.67 | 0.00 |

Fig. 3d where $R_t$ and $T_t$ are considered manipulative variables we need to intervene on to minimize $\log(U_t)$ at every time step. Time series data for 10 countries[5] are used to construct a non-parametric simulator and to compute the causal prior for both DCBO and CBO. DCBO convergences to the optimal intervention faster than competing approaches (see Table 1 and Fig. 6 in the appendix). The optimal sequence of interventions found in this experiment is equal to $\{(T_0, R_0) = (9.38, -2.00), (T_1, R_1) = (0.53, 6.00), (T_2) = (0.012)\}$ which is consistent with domain knowledge.

**Planktonic predator–prey community in a chemostat (ODE)** We investigate a biological system in which two species interact, one as a predator and the other as prey, with the goal of identifying the intervention reducing the concentration of dead animals in the chemostat – see $D_t$ in Fig. 3e. We use the system of ordinary differential equations (ODE) given by [6] as our SCM and construct the DAG by rolling out the temporal variable dependencies in the ODE while removing graph cycles. Observational data are provided in [6] and are use to compute the dynamic causal prior. DCBO outperforms competing methods in term of average gap metric and identifies the optimum faster (Table 1). Additional details can be found in the supplement (§6).

## 5 Conclusions

We consider the problem of finding a sequence of optimal interventions in a causal graph where causal temporal dependencies exist between variables. We propose the Dynamic Causal Bayesian Optimization (DCBO) algorithm which finds the optimal intervention at every time step by intervening in the system according to a causal acquisition function. Importantly, for each possible intervention we propose to use a surrogate model that incorporates information from previous interventions implemented in the system. This is constructed by exploiting theoretical results establishing the correlation structure among objective functions for two consecutive time steps as a function of the topology of the causal graph. We discuss the DCBO performance in a variety of setting characterized by different DAG properties and stationarity assumptions. Future work will focus on extending our theoretical results to more general DAG structures thus allowing for unobserved confounders and a changing DAG topology within each time step. In addition, we will work on combining the proposed framework with a causal discovery algorithm so as to account for uncertainty in the graph structure.

## Acknowledgements

This work was supported by the EPSRC grant EP/L016710/1, The Alan Turing Institute under EPSRC grant EP/N510129/1, the Defence and Security Programme at The Alan Turing Institute, funded by the UK Government and the Lloyds Register Foundation programme on Data Centric Engineering through the London Air Quality project. TD acknowledges support from UKRI Turing AI Fellowship (EP/V02678X/1).

---

[5]Data were downloaded from `https://www.data.oecd.org/` [Accessed: 01/04/2021]. All details in the supplement.

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
