# Supplementary material for
# Dynamic Causal Bayesian Optimisation

**Virginia Aglietti**[*]
University of Warwick
The Alan Turing Institute
V.Aglietti@warwick.ac.uk

**Neil Dhir**[*]
The Alan Turing Institute
ndhir@turing.ac.uk

**Javier González**
Microsoft Research Cambridge
Gonzalez.Javier@microsoft.com

**Theodoros Damoulas**
University of Warwick
The Alan Turing Institute
T.Damoulas@warwick.ac.uk

## Contents

[*]Denotes equal contribution

35th Conference on Neural Information Processing Systems (NeurIPS 2021).

# 1 Nomenclature

| Symbol | Description |
|---|---|
| $\mathbf{V}_t$ | Set of observable variables at time $t$ |
| $\mathbf{V}_{0:T}$ | Union of observable variables at time $t = 0, \ldots, T$ |
| $\mathbf{X}_t$ | Manipulative variables at time $t$ |
| $Y_t$ | Target variable at time $t$ |
| $\mathcal{P}(\mathbf{X}_t)$ | Power set of $\mathbf{X}_t$ |
| $\mathbb{M}_t$ | Set of MIS sets at time $t$ |
| $\mathbf{X}_{s,t}$ | $s$-th intervention set at time $t$ |
| $\mathcal{D}$ | Observational dataset $\{\mathbf{V}_{0:T}^i\}_{i=1}^N$ |
| $N$ | Number of observational data points collected from the system |
| $\mathcal{D}_{s,t}^I$ | Interventional data points collected for the intervention set $\mathbf{X}_{s,t}$ |
| $\mathbf{X}^I$ | Vector of interventional values |
| $\mathbf{Y}_{s,t}^I$ | Vector of target values obtained by intervening on $\mathbf{X}_{s,t}$ at $\mathbf{X}^I$ |
| $H$ | Maximum number of explorative interventions an agent can conduct at every $t$ |
| $I_{0:t-1}$ | Decision Interventions at time step $0$ to $t-1$ |
| $f_{s,t}$ | Objective function for the set $\mathbf{X}_{s,t}$ |
| $m_{s,t}$ | Prior mean function of GP on $f_{s,t}$ |
| $k_{s,t}$ | Prior kernel function of GP on $f_{s,t}$ |
| $m_{s,t}(\cdot \mid \mathcal{D}_{s,t}^I)$ | Posterior mean function for GP on $f_{s,t}$ |
| $k_{s,t}(\cdot, \cdot \mid \mathcal{D}_{s,t}^I)$ | Posterior covariance function for GP on $f_{s,t}$ |

## 2 Characterization of the time structure in a DAG with time dependent variables

In this section we give the proof for Theorem 1 in the main text. Consider the objective function $\mathbb{E}[Y_t|\,\mathrm{do}(\mathbf{X}_{s,t}=\mathbf{x}_{s,t})\,,I_{0:t-1}]$ and define the following sets:

- $\mathrm{Pa}(Y_t) = Y_t^{\mathrm{PT}} \cup Y_t^{\mathrm{PNT}}$ with $Y_t^{\mathrm{PT}} = \mathrm{Pa}(Y_t) \cap Y_{0:t-1}$ denoting the parents of $Y_t$ that are target variables at previous time steps and $Y_t^{\mathrm{PNT}} = \mathrm{Pa}(Y_t)\backslash Y_t^{\mathrm{PT}}$ including the parents of $Y_t$ that are not target variables.

- For any set $\mathbf{X}_{s,t} \in \mathcal{P}(\mathbf{X}_t)$, $\mathbf{X}_{s,t}^{\mathrm{PY}} = \mathbf{X}_{s,t} \cap \mathrm{Pa}(Y_t)$ includes the variables in $\mathbf{X}_{s,t}$ that are parents of $Y_t$ while $\mathbf{X}_{s,t}^{\mathrm{NPY}} = \mathbf{X}_{s,t}\backslash\mathbf{X}_{s,t}^{\mathrm{PY}}$ so that $\mathbf{X}_{s,t} = \mathbf{X}_{s,t}^{\mathrm{PY}} \cup \mathbf{X}_{s,t}^{\mathrm{NPY}}$.

- For any set $I_{0:t-1}^V \subseteq \mathbf{X}_{0:t-1}$, $I_{0:t-1}^{\mathrm{PY}} = I_{0:t-1}^V \cap \mathrm{Pa}(Y_t)$ includes the variables in $I_{0:t-1}^V$ that are parents of $Y_t$ and $I_{0:t-1}^{\mathrm{NPY}} = I_{0:t-1}^V\backslash I_{0:t-1}^{\mathrm{PY}}$ so that $I_{0:t-1}^V = I_{0:t-1}^{\mathrm{PY}} \cup I_{0:t-1}^{\mathrm{NPY}}$.

- For any two sets $\mathbf{X}_{s,t} \in \mathrm{Pa}(Y_t)$ and $I_{0:t-1}^V \subseteq \mathbf{X}_{0:t-1}$, $\mathbf{W}$ is a set such that $\mathrm{Pa}(Y_t) = Y_t^{\mathrm{PT}} \cup \mathbf{X}_{s,t}^{\mathrm{PY}} \cup I_{0:t-1}^{\mathrm{PY}} \cup \mathbf{W}$. This means that $\mathbf{W}$ includes those variables that are parents of $Y_t$ but are nor target at previous time steps nor intervened variables.

In the following proof the values of $I_{0:t-1}^V$, $\mathbf{X}_{s,t}^{\mathrm{PY}}$, $I_{0:t-1}^{\mathrm{PY}}$ and $W$ are denoted by $\mathbf{i}$, $\mathbf{x}^{\mathrm{PY}}$, $\mathbf{i}^{\mathrm{PY}}$ and $\mathbf{w}$ respectively. The values of $Y_t^{\mathrm{PT}}$, $\mathbf{X}_{s,t}^{\mathrm{NPY}}$ and $I_{0:t-1}^{\mathrm{NPY}}$ are instead represented by $\mathbf{y}_t^{\mathrm{PT}}$, $\mathbf{x}^{\mathrm{NPY}}$ and $\mathbf{i}^{\mathrm{NPY}}$. Finally, $f_Y^Y$ and $f_Y^{\mathrm{NY}}$ are the functions in the SCM for $Y_t$ (see Assumptions (1) in the main text).

***Proof of Theorem 1***  Under Assumptions 1 we can write :

$$\mathbb{E}[Y_t|\,\mathrm{do}(\mathbf{X}_{s,t}=\mathbf{x}_{s,t})\,,I_{0:t-1}] = \int y_t p(y_t|\,\mathrm{do}(\mathbf{X}_{s,t}=\mathbf{x}_{s,t})\,,I_{0:t-1})\mathrm{d}y_t$$

$$= \int \cdots \int y_t p\big(y_t|\,\mathrm{do}\big(\mathbf{X}_{s,t}^{\mathrm{PY}}=\mathbf{x}^{\mathrm{PY}}\big),\mathrm{do}\big(\mathbf{X}_{s,t}^{\mathrm{NPY}}=\mathbf{x}^{\mathrm{NPY}}\big),I_{0:t-1}^{\mathrm{PY}},I_{0:t-1}^{\mathrm{NPY}},\mathbf{y}_t^{\mathrm{PT}},\mathbf{w}\big)$$

$$\times p(\mathbf{y}_t^{\mathrm{PT}},\mathbf{w}|\,\mathrm{do}(\mathbf{X}_{s,t}=\mathbf{x}_{s,t})\,,I_{0:t-1})\mathrm{d}y_t\mathrm{d}\mathbf{y}_t^{\mathrm{PT}}\mathrm{d}\mathbf{w}$$

$$= \Big/\text{Rule 2 and Rule 1 of do-calculus}\Big/$$

$$= \int \cdots \int y_t p\big(y_t|\,\mathrm{do}\big(\mathbf{X}_{s,t}^{\mathrm{PY}}=\mathbf{x}^{\mathrm{PY}}\big),I_{0:t-1}^{\mathrm{PY}},\mathbf{y}_t^{\mathrm{PT}},\mathbf{w}\big)$$

$$\times p(\mathbf{y}_t^{\mathrm{PT}},\mathbf{w}|\,\mathrm{do}(\mathbf{X}_{s,t}=\mathbf{x}_{s,t})\,,I_{0:t-1})\mathrm{d}y_t\mathrm{d}\mathbf{y}_t^{\mathrm{PT}}\mathrm{d}\mathbf{w} \qquad (1)$$

$$= \int \cdots \int \mathbb{E}\big[Y_t|\,\mathrm{do}\big(\mathbf{X}_{s,t}^{\mathrm{PY}}=\mathbf{x}^{\mathrm{PY}}\big),I_{0:t-1}^{\mathrm{PY}},\mathbf{y}_t^{\mathrm{PT}},\mathbf{w}\big]$$

$$\times p(\mathbf{y}_t^{\mathrm{PT}},\mathbf{w}|\,\mathrm{do}(\mathbf{X}_{s,t}=\mathbf{x}_{s,t})\,,I_{0:t-1})\mathrm{d}\mathbf{y}_t^{\mathrm{PT}}\mathrm{d}\mathbf{w}$$

$$= \Big/\text{Assumption (2)}\Big/$$

$$= \int \cdots \int f_Y^Y(\mathbf{y}_t^{\mathrm{PT}}) + f_Y^{\mathrm{NY}}(\mathbf{x}^{\mathrm{PY}},\mathbf{i}^{\mathrm{PY}},\mathbf{w})$$

$$\times p(\mathbf{y}_t^{\mathrm{PT}},\mathbf{w}|\,\mathrm{do}(\mathbf{X}_{s,t}=\mathbf{x}_{s,t})\,,I_{0:t-1})\mathrm{d}\mathbf{y}_t^{\mathrm{PT}}\mathrm{d}\mathbf{w} \qquad (2)$$

$$= \int \cdots \int f_Y^Y(\mathbf{y}_t^{\mathrm{PT}})p(\mathbf{y}_t^{\mathrm{PT}},\mathbf{w}|\,\mathrm{do}(\mathbf{X}_{s,t}=\mathbf{x}_{s,t})\,,I_{0:t-1})\mathrm{d}\mathbf{y}_t^{\mathrm{PT}}\mathrm{d}\mathbf{w}$$

$$+ \int \cdots \int f_Y^{\mathrm{NY}}(\mathbf{x}^{\mathrm{PY}},\mathbf{i}^{\mathrm{PY}},\mathbf{w})p(\mathbf{y}_t^{\mathrm{PT}},\mathbf{w}|\,\mathrm{do}(\mathbf{X}_{s,t}=\mathbf{x}_{s,t})\,,I_{0:t-1})\mathrm{d}\mathbf{y}_t^{\mathrm{PT}}\mathrm{d}\mathbf{w}$$

$$= \int f_Y^Y(\mathbf{y}_t^{\mathrm{PT}}) p(\mathbf{y}_t^{\mathrm{PT}} | \operatorname{do}(\mathbf{X}_{s,t} = \mathbf{x}_{s,t}), I_{0:t-1}) d\mathbf{y}_t^{\mathrm{PT}} \tag{3}$$

$$+ \int f_Y^{\mathrm{NY}}(\mathbf{x}^{\mathrm{PY}}, \mathbf{i}^{\mathrm{PY}}, \mathbf{w}) p(\mathbf{w} | \operatorname{do}(\mathbf{X}_{s,t} = \mathbf{x}_{s,t}), I_{0:t-1}) d\mathbf{w}$$

$$= \Big/ \text{Time assumption} \Big/$$

$$= \int f_Y^Y(\mathbf{y}_t^{\mathrm{PT}}) p(\mathbf{y}_t^{\mathrm{PT}} | I_{0:t-1}) d\mathbf{y}_t^{\mathrm{PT}} + \int f_Y^{\mathrm{NY}}(\mathbf{x}^{\mathrm{PY}}, \mathbf{i}^{\mathrm{PY}}, \mathbf{w}) p(\mathbf{w} | \operatorname{do}(\mathbf{X}_{s,t} = \mathbf{x}_{s,t}), I_{0:t-1}) d\mathbf{w} \tag{4}$$

$$= \Big/ \text{Observed interventions} \Big/$$

$$= f_Y^Y(\mathbf{f}^\star) + \int f_Y^{\mathrm{NY}}(\mathbf{x}^{\mathrm{PY}}, \mathbf{i}^{\mathrm{PY}}, \mathbf{w}) p(\mathbf{w} | \operatorname{do}(\mathbf{X}_{s,t} = \mathbf{x}_{s,t}), I_{0:t-1}) d\mathbf{w} \tag{5}$$

$$= f_Y^Y(\mathbf{f}^\star) + \mathbb{E}_{p(\mathbf{w} | \operatorname{do}(\mathbf{X}_{s,t} = \mathbf{x}_{s,t}), I_{0:t-1})} \big[ f_Y^{\mathrm{NY}}(\mathbf{x}^{\mathrm{PY}}, \mathbf{i}^{\mathrm{PY}}, \mathbf{w}) \big] \tag{6}$$

with $\mathbf{f}^\star = \{ \mathbb{E}[Y_i | \operatorname{do}(\mathbf{X}_{s,i}^\star = \mathbf{x}_{s,i}^\star), I_{0:i-1}] \}_{Y_i \in Y_t^{\mathrm{PT}}}$ denoting the values of $Y_t^{\mathrm{PT}}$ corresponding to the optimal interventions implemented at previous time steps . Eq. (1) follows from $Y_t \perp\!\!\!\perp (\mathbf{X}_{s,t}^{\mathrm{NPY}} \cup I_{0:t-1}^{\mathrm{NPY}}) | \mathbf{X}_{s,t}^{\mathrm{PY}}, I_{0:t-1}^{\mathrm{PY}}, \mathbf{W}, Y_t^{\mathrm{PT}}$ in $\mathcal{G}_{\overline{\mathbf{X}_{s,t}^{\mathrm{PY}}, I_{0:t-1}^{\mathrm{PY}}} \underline{\mathbf{X}_{s,t}^{\mathrm{NPY}}, I_{0:t-1}^{\mathrm{NPY}}}}$ (Rule 2 of do-calculus) and $Y_t \perp\!\!\!\perp (\mathbf{X}_{s,t}^{\mathrm{NPY}} \cup I_{0:t-1}^{\mathrm{NPY}}) | \mathbf{X}_{s,t}^{\mathrm{PY}}, I_{0:t-1}^{\mathrm{PY}}, \mathbf{W}, Y_t^{\mathrm{PT}}$ in $\mathcal{G}_{\overline{\mathbf{X}_{s,t}^{\mathrm{PY}}, I_{0:t-1}^{\mathrm{PY}}}}$ (Rule 1 of do-calculus). Eq. (2) follows from the second assumption in Assumptions (1) in the main text. Eq. (4) follows from $Y_t^{\mathrm{PT}} \perp\!\!\!\perp \mathbf{X}_{s,t}$ as interventions at time $t$ cannot affect variables at time steps $0 : t-1$. Finally, noticing that $p(\mathbf{y}_t^{\mathrm{PT}} | I_{0:t-1})$ is the distribution targeted when optimizing the objective function at previous time steps one can obtain Eq. (6).

∎

The derivations above show how the objective function at time $t$ is given by the expected value of the output of the functional relationship $f_Y^{\mathrm{NY}}$ where the expectation is taken with respect to the variables that are not intervened on. This expectation is then shifted to account for the interventions implemented in the system at previous time steps that are affecting the target variable through $f_Y^Y$. Notice that, given our assumption on the absence of unobserved confounders, the distribution $p(\mathbf{w} | \operatorname{do}(\mathbf{X}_{s,t} = \mathbf{x}_{s,t}), I_{0:t-1})$ can be further simplified by conditioning on the variables in $\mathcal{G}$ that are on the back-door path between $(\mathbf{X}_{s,t}, I_{0:t-1})$ and $Y_t$ and are not colliders. When the variable $Y_t$ does not depend on the previous target nodes, the function $f_Y^Y$ does not exist and Eq. (6) reduces to

$$\mathbb{E}_{p(\mathbf{w} | \operatorname{do}(\mathbf{X}_{s,t} = \mathbf{x}_{s,t}), I_{0:t-1})} \big[ f_Y^{\mathrm{NY}}(\mathbf{x}^{\mathrm{PY}}, \mathbf{i}^{\mathrm{PY}}, \mathbf{w}) \big]. \tag{7}$$

In this case previous interventions impact the target variable at time $t$ by changing the distributions of the parents of $Y_t$ that are not intervened but the information in $\mathbf{f}^\star$ is lost.

Eq. (6) can be further manipulated to reduce the second term to a do-free expression. Instead of applying the rules of do-calculus, one can expand $p(\mathbf{w} | \operatorname{do}(\mathbf{X}_{s,t} = \mathbf{x}_{s,t}), I_{0:t-1})$ by further conditioning on the parents of $\mathbf{W}$ that are not in $(\mathbf{X}_{s,t} \cup I_{0:t-1})$. In this case, $\mathbf{w}$ in $f_Y^{\mathrm{NY}}(\mathbf{x}^{\mathrm{PY}}, \mathbf{i}^{\mathrm{PY}}, \mathbf{w})$ is replaced by the functions $\{ f_W(\cdot) \}_{W \in \mathbf{w}}$ in the SCM corresponding to the variables in $\mathbf{W}$ and computed in $\mathbf{w}$. This leads to a partial composition of $f_Y^{\mathrm{NY}}$ with $\{ f_W(\cdot) \}_{W \in \mathbf{w}}$ and can be repeated recursively until the set of variables with respect to which we are taking the expectation is a subset of $\mathbf{X}_{s,t}$ or $I_{0:t-1}^V$ thus making the distribution a delta function. For instance, when $\mathbf{W} \subset \mathbf{X}_{s,t}$ in Eq. (6), we have $p(\mathbf{w} | \operatorname{do}(\mathbf{X}_{s,t} = \mathbf{x}_{s,t}), I_{0:t-1}) = \delta(\mathbf{w} = \mathbf{x}^{\mathrm{W}})$ where $\mathbf{x}^{\mathrm{W}}$ are the values in $\mathbf{x}_{s,t}$ corresponding to the variables in $\mathbf{W}$. Therefore, Eq. (6) reduces to $f_Y^Y(\mathbf{f}^\star) + f_Y^{\mathrm{NY}}(\mathbf{x}^{\mathrm{PY}}, \mathbf{i}^{\mathrm{PY}}, \mathbf{x}^{\mathrm{W}})$.

For a generic $W \in \mathbf{W} \not\subseteq (\mathbf{X}_{s,t} \cup I_{0:t-1}^V)$, denote by $\mathbf{X}_{s,t}^{\mathrm{PW}}$ and $I_{0:t-1}^{\mathrm{PW}}$ the subset of variables in $\mathbf{X}_{s,t}$ and $I_{0:t-1}$ that are parents of $W$ with corresponding values $\mathbf{x}^{\mathrm{PW}}$ and $\mathbf{i}^{\mathrm{PW}}$. Let $R = \operatorname{Pa}(W) \setminus (\mathbf{X}_{s,t}^{\mathrm{PW}} \cup I_{0:t-1}^{\mathrm{PW}})$ and $r$ be the corresponding value. We can define the $C(\cdot)$ function as:

$$C(W) = \begin{cases} f_W(\mathbf{u}_W, \mathbf{x}^{\mathrm{PW}}, \mathbf{i}^{\mathrm{PW}}) & \text{if } R = \varnothing \\ f_W(\mathbf{u}_W, \mathbf{x}^{\mathrm{PW}}, \mathbf{i}^{\mathrm{PW}}, r) & \text{if } R \subseteq \mathbf{X}_{s,t} \cup I_{0:t-1}^V \\ f_W(\mathbf{u}_W, \mathbf{x}^{\mathrm{PW}}, \mathbf{i}^{\mathrm{PW}}, C(R)) & \text{if } R \not\subseteq \mathbf{X}_{s,t} \cup I_{0:t-1}^V \end{cases} \tag{8}$$

with $u_W$ representing the exogenous variables with edges into $W$ and $f_W$ denoting the functional mapping for $W$ in the SCM. Note that if $R = \varnothing$ and $\mathbf{X}_{s,t}^{\mathrm{PW}}$ and $I_{0:t-1}^{\mathrm{PW}}$ are also empty then $f_W(\mathbf{u}_W, \mathbf{x}^{\mathrm{PW}}, \mathbf{i}^{\mathrm{PW}})$ reduces to $f_W(\mathbf{u}_W)$. The same holds for the other cases that is $f_W(\mathbf{u}_W, \mathbf{x}^{\mathrm{PW}}, \mathbf{i}^{\mathrm{PW}}, r) = f_W(\mathbf{u}_W, r)$ and $f_W(\mathbf{u}_W, \mathbf{x}^{\mathrm{PW}}, \mathbf{i}^{\mathrm{PW}}, C(R)) = f_W(\mathbf{u}_W, C(R))$ when $\mathbf{X}_{s,t}^{\mathrm{PW}}, I_{0:t-1}^{\mathrm{PW}} = \varnothing$. Exploiting Eq. (8) we can rewrite Eq. (6) as:

$$\mathbb{E}[Y_t \,|\, \mathrm{do}(\mathbf{X}_{s,t} = \mathbf{x}_{s,t})\,, I_{0:t-1}] = f_Y^Y(\mathbf{f}^\star) + \mathbb{E}_{p(\mathbf{U}_{0:t})}\big[f_Y^{\mathrm{NY}}(\mathbf{x}^{\mathrm{PY}}, \mathbf{i}^{\mathrm{PY}}, \{C(W)\}_{W \in \mathbf{w}})\big] \qquad (9)$$

The distribution $p(\mathbf{U}_{0:t})$ can be further simplified to consider only the exogenous variables with outgoing edges into the variables on the directed paths between $\mathbf{X}_{s,t}$ and $Y_t^{\mathrm{PNT}}$ and between $I_{0:t-1}^V$ and $Y_t^{\mathrm{PNT}}$. Notice how the second term in Eq. (9) propagates the interventions, both at the present and past time steps, through the SCM so as to express the parents of the target variable as a function of the intervened values. The expected target is then obtained as the propagation of the intervened variables and intervened targets through the function $f_{Y_t}$ in the SCM.

# 3 Example derivations

Next we show how one can use Theorem 1 to derive some of the objective functions used by DCBO for the DAGs in Fig. 1.

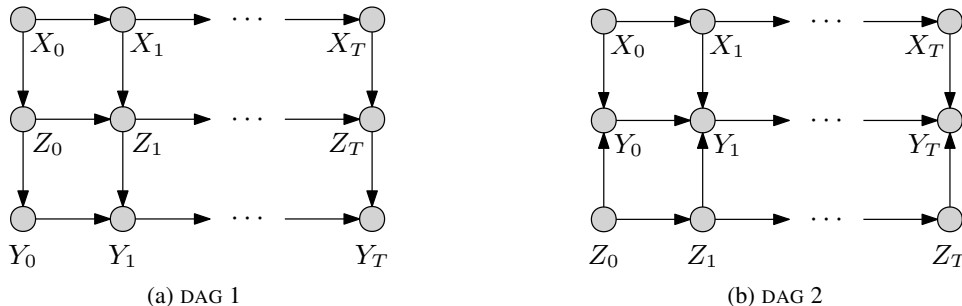

(a) DAG 1          (b) DAG 2

Figure 1: Dynamic Bayesian networks with different topologies. Figure 1a shows a DAG in which (per time-slice) the manipulative variable $X$ flows through $Z$, whereas in Fig. 1b the manipulative variables are independent of each other (note the direction of the vertical edges).

## 3.1 Derivations for DAG 1 in Fig. 1a

Consider the DAG in Fig. 1a and assume that the optimal intervention implemented at time $t = 0$ is given by $I_0 = \mathrm{do}(Z_0 = z_0^\star)$ and gives a target value of $y_0^\star$. At $t = 1$ the target variable is $Y_1$, $Y_t^{\mathrm{PT}} = \{Y_0\}$ and $Y_t^{\mathrm{PNT}} = \{Z_1\}$. Given $I_0$ we have $I_{0:t-1}^{\mathrm{PY}} = \varnothing$ and $I_{0:t-1}^{\mathrm{NPY}} = Z_0$. We can write the objective functions by noticing that, for $\mathbf{X}_{s,1} = \{Z_1\}$ we have $\mathbf{X}_{s,t}^{\mathrm{PY}} = \{Z_1\}$, $\mathbf{X}_{s,t}^{\mathrm{NPY}} = \varnothing$ and $W = \varnothing$, while for $\mathbf{X}_{s,1} = \{X_1\}$ we have $\mathbf{X}_{s,t}^{\mathrm{PY}} = \varnothing$, $\mathbf{X}_{s,t}^{\mathrm{NPY}} = \{X_1\}$ and $W = \{Z_1\}$. We do not compute the objective function for $\mathbf{X}_{s,1} = \{X_1, Z_1\}$ as this is equal to the function for $\mathbf{X}_{s,1} = \{Z_1\}$. Starting with $\mathbf{X}_{s,1} = \{Z_1\}$ we have:

$$
\begin{aligned}
\mathbb{E}[Y_1 \mid \mathrm{do}(Z_1 = z), I_0] &= \int y_1 p(y_1 \mid \mathrm{do}(Z_1 = z), I_0) \mathrm{d}y_1 \\
&= \int \int y_1 p(y_1 \mid y_0, \mathrm{do}(Z_1 = z), I_0) p(y_0 \mid \mathrm{do}(Z_1 = z), I_0) \mathrm{d}y_1 \mathrm{d}y_0 \\
&= \int \mathbb{E}[Y_1 \mid y_0, \mathrm{do}(Z_1 = z)] p(y_0 \mid \mathrm{do}(Z_1 = z), I_0) \mathrm{d}y_0 \\
&= \int [f_Y^Y(y_0) + f_Y^{\mathrm{NY}}(z)] p(y_0 \mid I_0) \mathrm{d}y_0 \\
&= \int f_Y^Y(y_0) p(y_0 \mid I_0) \mathrm{d}y_0 + f_Y^{\mathrm{NY}}(z) \\
&= f_Y^Y(y_0^\star) + f_Y^{\mathrm{NY}}(z)
\end{aligned}
$$

Notice that here $\mathbf{X}_{s,t}^{\mathrm{PY}} = \{Z_1\}$, $I_{0:t-1}^{\mathrm{PY}} = \varnothing$ and $W = \varnothing$ thus $\mathbb{E}_{p(\mathbf{w} \mid \mathrm{do}(\mathbf{X}_{s,t} = \mathbf{x}_{s,t}), I_{0:t-1})}\left[f_Y^{\mathrm{NY}}(\mathbf{x}^{\mathrm{PY}}, \mathbf{i}^{\mathrm{PY}}, \mathbf{w})\right] = f_Y^{\mathrm{NY}}(z)$. The objective function for $\mathbf{X}_{s,1} = \{X_1\}$ can be written as:

$$\mathbb{E}[Y_1|\operatorname{do}(X_1 = x), I_0] = \int y_1 p(y_1|\operatorname{do}(X_1 = x), I_0) \mathrm{d}y_1$$

$$= \int \int \int y_1 p(y_1|y_0, z_1, \operatorname{do}(X_1 = x), I_0) p(y_0, z_1|\operatorname{do}(X_1 = x), I_0) \mathrm{d}y_1 \mathrm{d}y_0 \mathrm{d}z_1$$

$$= \int \int \int y_1 p(y_1|y_0, z_1) p(y_0, z_1|\operatorname{do}(X_1 = x), I_0) \mathrm{d}y_1 \mathrm{d}y_0 \mathrm{d}z_1$$

$$= \int \int \mathbb{E}[Y_1|y_0, z_1] p(y_0, z_1|\operatorname{do}(X_1 = x), I_0) \mathrm{d}y_0 \mathrm{d}z_1$$

$$= \int \int [f_Y^Y(y_0) + f_Y^{\mathrm{NY}}(z_1)] p(y_0, z_1|\operatorname{do}(X_1 = x), I_0) \mathrm{d}y_0 \mathrm{d}z_1$$

$$= \int \int f_Y^Y(y_0) p(y_0, z_1|\operatorname{do}(X_1 = x), I_0) \mathrm{d}y_0 \mathrm{d}z_1 \tag{10}$$

$$+ \int \int f_Y^{\mathrm{NY}}(z_1) p(y_0, z_1|\operatorname{do}(X_1 = x), I_0) \mathrm{d}y_0 \mathrm{d}z_1$$

$$= \int f_Y^Y(y_0) p(y_0|I_0) \mathrm{d}y_0 + \int \int f_Y^{\mathrm{NY}}(z_1) p(z_1|\operatorname{do}(X_1 = x), I_0) \mathrm{d}z_1$$

$$= f_Y^Y(y_0^\star) + \int f_Y^{\mathrm{NY}}(z_1) p(z_1|\operatorname{do}(X_1 = x), I_0) \mathrm{d}z_1 \tag{11}$$

In this case $\mathbf{X}_{s,t}^{\mathrm{PY}} = \varnothing$, $I_{0:t-1}^{\mathrm{PY}} = \varnothing$ and $\mathbf{W} = \{Z_1\}$ thus

$$\mathbb{E}_{p(\mathbf{w}|\operatorname{do}(\mathbf{X}_{s,t} = \mathbf{x}_{s,t}), I_{0:t-1})} \left[ f_Y^{\mathrm{NY}}(\mathbf{x}^{\mathrm{PY}}, \mathbf{i}^{\mathrm{PY}}, \mathbf{w}) \right] = \mathbb{E}_{p(z_1|\operatorname{do}(X_1 = x), I_0)} \left[ f_Y^{\mathrm{NY}}(z_1) \right]. \tag{12}$$

We can further expand Eq. (11) noticing that in this case $\mathbf{W} = \{Z_1\} \not\subseteq \{X_1, Z_0\}$ but $\mathbf{X}_{s,t}^{PW} = \{X_1\}$, $I_{0:t-1}^{PW} = \{Z_0\}$ and $R = \varnothing$. Therefore we have $C(Z_1) = f_{Z_1}(\epsilon_{Z_1}, x_1, z_1)$ and Eq. (11) becomes:

$$\mathbb{E}[Y|\operatorname{do}(X_1 = x), I_0] = f_Y^Y(y_0^\star) + \int f_Y^{\mathrm{NY}}(z_1) p(z_1|\operatorname{do}(X_1 = x), I_0) \mathrm{d}z_1$$

$$= f_Y^Y(y_0^\star) + \int \int f_Y^{\mathrm{NY}}(z_1) p(z_1|\epsilon_{Z_1}, \operatorname{do}(X_1 = x), I_0) p(\epsilon_{Z_1}|\operatorname{do}(X_1 = x), I_0) \mathrm{d}z_1 \mathrm{d}\epsilon_{Z_1}$$

$$= f_Y^Y(y_0^\star) + \int \int f_Y^{\mathrm{NY}}(z_1) \delta(z_1 = f_{Z_1}(\epsilon_{Z_1}, x, z_0^\star)) p(\epsilon_{Z_1}) \mathrm{d}z_1 \mathrm{d}\epsilon_{Z_1}$$

$$= f_Y^Y(y_0^\star) + \mathbb{E}_{p(\epsilon_{Z_1})} \left[ f_Y^{\mathrm{NY}}(f_{Z_1}(\epsilon_{Z_1}, x, z_0^\star)) \right].$$

### 3.2 Derivations for DAG 2 in Fig. 1b

Next we consider the DAG in Fig. 1b and assume that the optimal interventions implemented at time $t = 0$ and $t = 1$ are given by $I_0 = \operatorname{do}(X_0 = x_0^\star)$ and $I_1 = \operatorname{do}(Z_1 = z_1^\star)$. The optimal target values associated with these two interventions are given by $y_0^\star$ and $y_1^\star$ respectively. We are interested in computing two objective functions: $\mathbb{E}[Y_2|\operatorname{do}(X_2 = x_2), I_0, I_1]$ and $\mathbb{E}[Y_2|\operatorname{do}(Z_2 = z_2), I_0, I_1]$. In this case $\mathbf{y}_t^{\mathrm{PT}} = \{Y_1\}$, $Y_t^{\mathrm{PNT}} = \{X_2, Z_2\}$, $I_{0:t-1}^{\mathrm{PY}} = \varnothing$ and $I_{0:t-1}^{\mathrm{NPY}} = \{X_0, Z_1\}$. Starting from $\mathbb{E}[Y_2|\operatorname{do}(X_2 = x_2), I_0, I_1]$, when $\mathbf{X}_{s,2} = \{X_2\}$ we have $\mathbf{X}_{s,t}^{\mathrm{PY}} = \{X_2\}$, $\mathbf{X}_{s,t}^{\mathrm{NPY}} = \varnothing$ and $W = \{Z_2\}$. We can write:

$$\mathbb{E}[Y_2|\operatorname{do}(X_2=x_2),I_0,I_1]=\int y_2 p(y_2|\operatorname{do}(X_2=x_2),I_0,I_1)\mathrm{d}y_2$$

$$=\int\int\int y_2 p(y_2|y_1,z_2,\operatorname{do}(X_2=x_2),I_0,I_1)p(y_1,z_2|\operatorname{do}(X_2=x_2),I_0,I_1)\mathrm{d}y_2\mathrm{d}y_1\mathrm{d}z_2$$

$$=\int\int\int y_2 p(y_2|y_1,z_2,\operatorname{do}(X_2=x_2))p(y_1,z_2|\operatorname{do}(X_2=x_2),I_0,I_1)\mathrm{d}y_2\mathrm{d}y_1\mathrm{d}z_2$$

$$=\int\int\mathbb{E}[Y_2|y_1,z_2,\operatorname{do}(X_2=x_2)]p(y_1,z_2|\operatorname{do}(X_2=x_2),I_0,I_1)\mathrm{d}y_1\mathrm{d}z_2$$

$$=\int\int[f_Y^Y(y_1)+f_Y^{\mathrm{NY}}(x_2,z_2)]p(y_1,z_2|\operatorname{do}(X_2=x_2),I_0,I_1)\mathrm{d}y_1\mathrm{d}z_2$$

$$=\int\int f_Y^Y(y_1)p(y_1,z_2|\operatorname{do}(X_2=x_2),I_0,I_1)\mathrm{d}y_1\mathrm{d}z_2$$

$$+\int\int f_Y^{\mathrm{NY}}(x_2,z_2)p(y_1,z_2|\operatorname{do}(X_2=x_2),I_0,I_1)\mathrm{d}y_1\mathrm{d}z_2$$

$$=\int f_Y^Y(y_1)p(y_1|I_0,I_1)\mathrm{d}y_1+\int f_Y^{\mathrm{NY}}(x_2,z_2)p(z_2|\operatorname{do}(X_2=x_2),I_0,I_1)\mathrm{d}z_2$$

$$=f_Y^Y(y_1^\star)+\int f_Y^{\mathrm{NY}}(x_2,z_2)p(z_2|I_1)\mathrm{d}z_2$$

$$=f_Y^Y(y_1^\star)+\mathbb{E}_{p(\epsilon_{Z_2})}\big[f_Y^{\mathrm{NY}}(x_2,f_{Z_2}(z_1^\star,\epsilon_{Z_2}))\big]$$

Next we compute $\mathbb{E}[Y_2|\operatorname{do}(Z_2=z_2),I_0,I_1]$ by noticing that, when $\mathbf{X}_{s,2}=\{Z_2\}$, we have $\mathbf{X}_{s,t}^{\mathrm{PY}}=\{Z_2\}$, $\mathbf{X}_{s,t}^{\mathrm{NPY}}=\varnothing$ and $W=\{X_2\}$. In this case we have:

$$\mathbb{E}[Y_2|\operatorname{do}(Z_2=z_2),I_0,I_1]=\int y_2 p(y_2|\operatorname{do}(Z_2=z_2),I_0,I_1)\mathrm{d}y_2$$

$$=\int\int\int y_2 p(y_2|y_1,x_2,\operatorname{do}(Z_2=z_2),I_0,I_1)p(y_1,x_2|\operatorname{do}(Z_2=z_2),I_0,I_1)\mathrm{d}y_2\mathrm{d}y_1\mathrm{d}x_2$$

$$=\int\int\int y_2 p(y_2|y_1,x_2,\operatorname{do}(Z_2=z_2))p(y_1,x_2|\operatorname{do}(Z_2=z_2),I_0,I_1)\mathrm{d}y_2\mathrm{d}y_1\mathrm{d}x_2$$

$$=\int\int\mathbb{E}[Y_2|y_1,x_2,\operatorname{do}(Z_2=z_2)]p(y_1,x_2|\operatorname{do}(Z_2=z_2),I_0,I_1)\mathrm{d}y_1\mathrm{d}x_2$$

$$=\int\int[f_Y^Y(y_1)+f_Y^{\mathrm{NY}}(x_2,z_2)]p(y_1,x_2|\operatorname{do}(Z_2=z_2),I_0,I_1)\mathrm{d}y_1\mathrm{d}x_2$$

$$=\int f_Y^Y(y_1)p(y_1|I_0,I_1)\mathrm{d}y_1+\int f_Y^{\mathrm{NY}}(x_2,z_2)p(x_2|\operatorname{do}(Z_2=z_2),I_0,I_1)\mathrm{d}x_2$$

$$=f_Y^Y(y_1^\star)+\int f_Y^{\mathrm{NY}}(x_2,z_2)p(x_2|\operatorname{do}(Z_2=z_2),I_0,I_1)\mathrm{d}x_2\qquad(13)$$

Let's now focus on Eq. (13). Here $\mathbf{W}=\{X_2\}\not\subseteq\{Z_2,X_0,Z_1\}$, $\mathbf{X}_{s,t}^{\mathrm{PW}}=\varnothing$, $I_{0:t-1}^{\mathrm{PW}}=\varnothing$ and $R=\{X_1\}$. Therefore we have $C(X_2)=f_{X_2}(\epsilon_{X_2},C(R))$ as $R\not\subseteq\{Z_2,X_0,Z_1\}$. We thus need to compute $C(R)=C(X_1)$. When $W=X_1$, $\mathbf{X}_{s,t}^{\mathrm{PW}}=\varnothing$ but $I_{0:t-1}^{\mathrm{PW}}=\{X_0\}$ and $R=\varnothing$. We can thus write $C(X_2)=f_{X_2}(\epsilon_{X_2},f_{X_1}(\epsilon_{X_1},x_0))$ and replace it in Eq. (13) to get:

$$\mathbb{E}[Y_2|\operatorname{do}(Z_2=z_2),I_0,I_1]=f_Y^Y(y_1^\star)+\mathbb{E}_{p(\epsilon_{X_2})p(\epsilon_{X_1})}\big[f_Y^{\mathrm{NY}}(f_{X_2}(\epsilon_{X_2},f_{X_1}(\epsilon_{X_1},x_0)),z_2)\big].$$

# 4 Reducing the search space

In this section we give the proof for Proposition 3.1 in the main text. Denote by $\mathbb{M}_t \subseteq \mathcal{P}(\mathbf{X}_t)$ the set of MISs at time $t$ and let $\mathbb{S}_t = \mathcal{P}(\mathbf{X}_t) \backslash \mathbb{M}_t$ include the sets that are not MIS. For any set $\mathbf{X}_{s,t} \in \mathbb{S}_t$ we denote the *superfluous* variables by $\mathbf{S}_{s,t}$. These are the variables not needed in the computation of the objective functions that is those variables for which $\mathbb{E}[Y_t|\operatorname{do}(\mathbf{X}_{s,t} = \mathbf{x}_{s,t}), I_{0:t-1}] = \mathbb{E}[Y_t|\operatorname{do}(\mathbf{X}'_{s,t} = \mathbf{x}'_{s,t}), I_{0:t-1}]$ where $\mathbf{X}'_{s,t} = \mathbf{X}_t \backslash \mathbf{S}_{s,t}$. Given the initial set of MISs at time $t = 0$ represented by $\mathbb{M}_0$ we have:

**Proposition 4.1. Minimal intervention sets in time.** If $\mathcal{G}_t = \mathcal{G}, \forall t$ then $\mathbb{M}_t = \mathbb{M}_0$ for $t > 0$.

*Proof.* Consider a generic set $\mathbf{X}_{s,t} \in \mathbb{S}_t$. The corresponding objective function can be written as:

$$
\begin{aligned}
\mathbb{E}[Y_t|\operatorname{do}(\mathbf{X}_{s,t} = \mathbf{x}_{s,t}), I_{0:t-1}] &= \mathbb{E}[Y_t|\operatorname{do}(\mathbf{X}'_{s,t} = \mathbf{x}'_{s,t}), \operatorname{do}(\mathbf{S}_{s,t} = \mathbf{s}_{s,t}), I_{0:t-1}] \\
&= \int \mathbb{E}[Y_t|\operatorname{do}(\mathbf{X}'_{s,t} = \mathbf{x}'_{s,t}), \operatorname{do}(\mathbf{S}_{s,t} = \mathbf{s}_{s,t}), I_{0:t-1}, \mathbf{V}_{0:t-1} \backslash I_{0:t-1}] \\
&\quad \times p(\mathbf{V}_{0:t-1} \backslash I_{0:t-1}|\operatorname{do}(\mathbf{X}'_{s,t} = \mathbf{x}'_{s,t}), \operatorname{do}(\mathbf{S}_{s,t} = \mathbf{s}_{s,t}), I_{0:t-1})\operatorname{d}\mathbf{V}_{0:t-1} \\
&= \int \mathbb{E}[Y_t|\operatorname{do}(\mathbf{X}'_{s,t} = \mathbf{x}'_{s,t}), I_{0:t-1}, \mathbf{V}_{0:t-1} \backslash I_{0:t-1}] \qquad (14) \\
&\quad \times p(\mathbf{V}_{0:t-1} \backslash I_{0:t-1}|\operatorname{do}(\mathbf{X}'_{s,t} = \mathbf{x}'_{s,t}), I_{0:t-1})\operatorname{d}\mathbf{V}_{0:t-1} \\
&= \mathbb{E}[Y_t|\operatorname{do}(\mathbf{X}'_{s,t} = \mathbf{x}'_{s,t}), I_{0:t-1}] \qquad (15)
\end{aligned}
$$

where Eq. (14) can be obtained by noticing that $Y_t \perp\!\!\!\perp \mathbf{S}_{s,t}|\mathbf{X}'_{s,t}, I_{0:t-1}, \mathbf{V}_{0:t-1} \backslash I_{0:t-1}$ in $\mathcal{G}_{\overline{\mathbf{S}_{s,t}, I_{0:t-1}, \mathbf{X}'_{s,t}}}$. This is due to the fact that $\mathbf{S}_{s,t}$ does not have back door paths to $Y_t$ in $\mathcal{G}_{\overline{\mathbf{S}_{s,t}, I_{0:t-1}, \mathbf{X}'_{s,t}}}$ and its front door paths to $Y_t$ in $\mathcal{G}_{\overline{\mathbf{S}_{s,t}, I_{0:t-1}, \mathbf{X}'_{s,t}}}$ are blocked by $\mathbf{X}'_{s,t}$. Indeed, $\mathbf{S}_{s,t}$ cannot have outgoing edges to variables in $0 : t - 1$ and the front door paths to $Y_t$ going through variables at time $t$ are blocked by definition of a MIS set by $\mathbf{X}'_{s,t}$ in $\mathcal{G}_t = \mathcal{G}, \forall t$. $\square$

# 5 Additional experimental details and results

This section contains additional experimental details associated to the experiments discussed in Section 4 of the main text.

## 5.1 Stationary DAG and SCM (STAT.)

The SCM used for this experiments is given by:

$$X_t = X_{t-1}\mathbb{1}_{t>0} + \epsilon_X$$
$$Z_t = \exp(-X_t) + Z_{t-1}\mathbb{1}_{t>0} + \epsilon_Z$$
$$Y_t = \cos(Z_t) - \exp(-Z_t/20) + Y_{t-1}\mathbb{1}_{t>0} + \epsilon_Y$$

where $\epsilon_i \sim \mathcal{N}(0,1)$ for $i \in \{X, Z, Y\}$ and $\mathbb{1}_{t>0}$ represent an indicator function that is equal to one $t > 0$ and zero otherwise. We run this experiment 10 times by setting $T = 3$, $N = 10$, $D(X_t) = \{-5.0, 5.0\}$ and $D(Z_t) = \{-5.0, 20.0\}$. Notice that given the DAG (Fig. X) we have $\mathbb{M}_t = \{\{X_t\}, \{Z_t\}\}$.

The right panel of Fig. 2 shows the true objective functions together with the optimal intervention per time step (1st row), the dynamic causal GP model for the intervention on $Z$ (2nd row) and the convergence of the DCBO algorithm to the optimum (3rd row). Notice how the location of the optimum changes significantly both in terms of optimal set and intervention value when going from $t = 0$ to $t = 1$. DCBO quickly identifies the optimum via the prior dependency on $y^\star_{0:t-1}$.

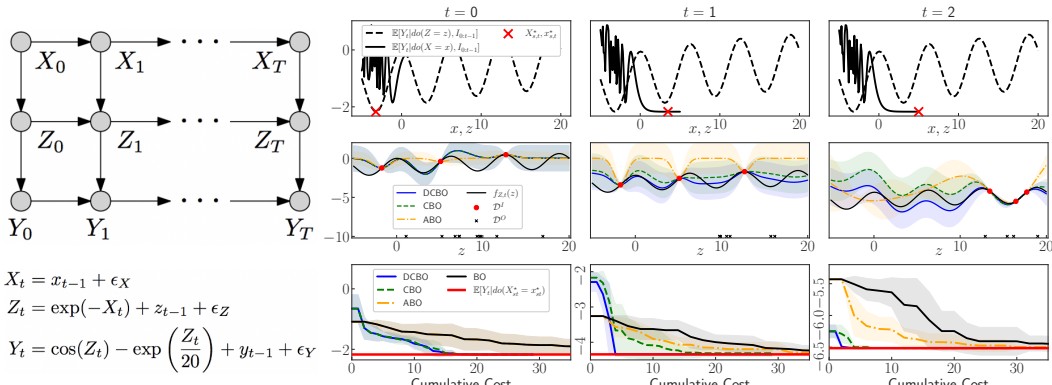

Figure 2: Stationary synthetic experiment (STAT.). *Left panel*: $\mathcal{G}_{0:T}$ and SCM. *Right panel, 1st row*: Objective functions for the sets in $\mathbb{M} = \{\{Z\}, \{X\}\}$. *Right panel, 2nd row*: Posterior GP obtained when using the dynamic causal GP construction vs alternative models. *Right panel, 3rd row*: Convergence of DCBO and alternative models to the true optimum (red line) across 10 replicates. Shaded areas give $\pm$ one standard deviation.

## 5.2 Noisy manipulative variables (NOISY)

The SCM used for this experiments is given by:

$$X_t = X_{t-1}\mathbb{1}_{t>0} + \epsilon_X$$
$$Z_t = \exp(-X_t) + Z_{t-1}\mathbb{1}_{t>0} + \epsilon_Z$$
$$Y_t = \cos(Z_t) - \exp(-Z_t/20) + Y_{t-1}\mathbb{1}_{t>0} + \epsilon_Y$$

where, differently from before, we have $\epsilon_Y \sim \mathcal{N}(0,1)$ and $\epsilon_i \sim \mathcal{N}(2,4)$ for $i \in \{X, Z\}$. We keep the remaining parameters equal to the previous experiment. This means $T = 3$, $N = 10$, $D(X_t) = \{-5.0, 5.0\}$ and $D(Z_t) = \{-5.0, 20.0\}$.

## 5.3 Missing observational data (MISS.)

For this experiment we use the same SCM of the experiment STAT. However, we set $T = 6$, $N = 10$ for the first three time steps and $N = 0$ afterwards. Fig. 3 shows the convergence paths for this

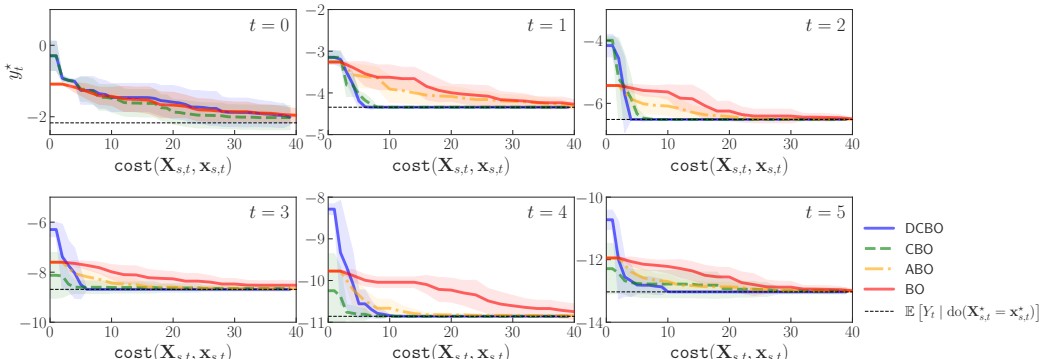

Figure 3: Experiment MISS.. Convergence of DCBO and competing methods across replicates. The red line gives the optimal $y_t^*, \forall t$. Shaded areas are $\pm$ standard deviation.

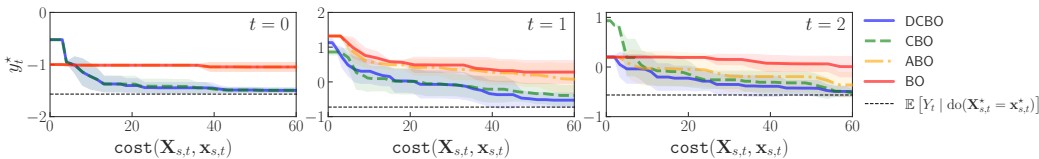

Figure 4: Experiment MULTIV.. Convergence of DCBO and competing methods across replicates. The red line gives the optimal $y_t^*, \forall t$. Shaded areas are $\pm$ standard deviation.

experiment. In this setting DCBO consistently outperform CBO at every time step. However, notice how ABO performance improves over time and outperforms DCBO starting from $t = 5$. This is due to the ability of ABO to learn the time dynamic of the objective function and exploit all interventional data collected over time to predict at the next time step.

## 5.4 Multivariate intervention sets (MULTIV.)

The SCM used for this experiments is given by:

$$
\begin{aligned}
W_t &= \epsilon_W \\
X_t &= -X_{t-1}\mathbb{1}_{t>0} + \epsilon_X \\
Z_t &= \sin(W_t) - Z_{t-1}\mathbb{1}_{t>0} + \epsilon_Z \\
Y_t &= -2 * \exp(-(X_t - 1)^2) - \exp(-(X_t + 1)^2) - (Z_t - 1)^2 \\
&\quad - Z_T^2 + \cos(Z_t * Y_{t-1}) - Y_{t-1}\mathbb{1}_{t>0} + \epsilon_Y
\end{aligned}
$$

where $\epsilon_i \sim \mathcal{N}(0, 1)$ for $i \in \{X, Z, W, Y\}$. We set $T = 3$, $N = 500$, $D(X_t) = \{-5.0, 5.0\}$, $D(Z_t) = \{-5.0, 20.0\}$ and $D(W_t) = \{-3.0, 3.0\}$. Notice that here DCBO and CBO explore the set $\mathbb{M}_t = \{\{X_t\}, \{Z_t\}, \{X_t, Z_t\}\}$ while BO and ABO intervene on $\{X_t, Z_t, W_t\}$. Fig. 4 shows the convergence paths for this experiment.

## 5.5 Independent manipulative variables (IND.)

The SCM used for this experiments is given by:

$$
\begin{aligned}
X_t &= -X_{t-1}\mathbb{1}_{t>0} + \epsilon_X \\
Z_t &= -Z_{t-1}\mathbb{1}_{t>0} + \epsilon_Z \\
Y_t &= -2 * \exp(-(X_t - 1)^2) - \exp(-(X_t + 1)^2) - (Z_t - 1)^2 \\
&\quad - Z_T^2 + \cos(Z_t * Y_{t-1}) - Y_{t-1}\mathbb{1}_{t>0} + \epsilon_Y
\end{aligned}
$$

where $\epsilon_i \sim \mathcal{N}(0, 1)$ for $i \in \{X, Z, Y\}$. We set $T = 3$, $N = 10$, $D(X_t) = \{-5.0, 5.0\}$ and $D(Z_t) = \{-5.0, 20.0\}$. Notice that here DCBO and CBO explore the set $\mathbb{M}_t = \{\{X_t\}, \{Z_t\}, \{X_t, Z_t\}\}$ while BO and ABO intervene on $\{X_t, Z_t\}$. In this case, exploring $\mathbb{M}_t$ and propagating uncertainty in the causal prior slows down DCBO convergence, see Fig. 5.

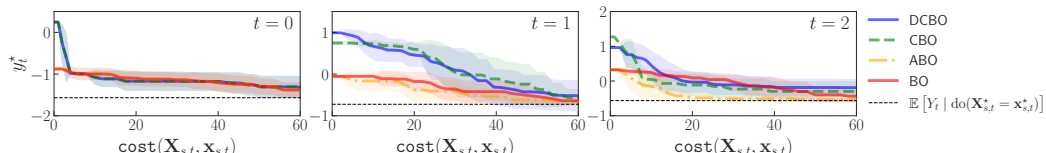

Figure 5: Experiment IND. Convergence of DCBO and competing methods across replicates. The red line gives the optimal $y_t^*$, $\forall t$. Shaded areas are $\pm$ standard deviation.

## 5.6 Non-stationary DAG and SCM (NONSTAT.)

The SCM used for this experiment is more complex than the others due to the fact that the DAG is non-stationary but so too is the SCM:

$$
\begin{cases}
f(t) & \text{if } t = 0 \\
g(t) & \text{if } t = 1 \\
h(t) & \text{if } t = 2
\end{cases}
\tag{16}
$$

where

$$
f(t) = \begin{cases}
X_t & = \epsilon_X \\
Z_t & = X_t + \epsilon_Z \\
Y_t & = \sqrt{|36 - (Z_t - 1)^2|} + 1 + \epsilon_Y
\end{cases}
$$

$$
g(t) = \begin{cases}
X_t & = X_{t-1} + \epsilon_X \\
Z_t & = -\frac{X_t}{X_{t-1}} + Z_{t-1} + \epsilon_Z \\
Y_t & = Z_t \cos(Z_t \pi) - Y_{t-1} + \epsilon_Y
\end{cases}
$$

$$
h(t) = \begin{cases}
X_t & = X_{t-1} + \epsilon_X \\
Z_t & = X_t + Z_{t-1} + \epsilon_Z \\
Y_t & = Z_t - Y_{t-1} - Z_{t-1} + \epsilon_Y
\end{cases}
$$

where $\epsilon_i \sim \mathcal{N}(0,1)$ for $i \in \{X, Z, Y\}$. We set $T = 3$, $N = 10$, $D(X_t) = \{-5.0, 5.0\}$ and $D(Z_t) = \{-5.0, 20.0\}$. Notice that here DCBO and CBO explore the set $\mathbb{M}_t = \{\{X_t\}, \{Z_t\}, \{X_t, Z_t\}\}$ while BO and ABO intervene on $\{X_t, Z_t\}$.

## 5.7 Real-World Economic data (ECON.)

We create an observational data set by extracting the following indicators from the OECD data portal (https://data.oecd.org/):

- GDP = GDP in milion of US dollars.
- CPI = annual growth of inflation measured by consumer price index CPI.
- TAXREV = tax revenues measured as a percentage of GDP.
- HUR = unemployment rate as measured by the numbers of unemployed people as a percentage of the labour force.

We manipulate these indicators to get the nodes in the DAG of Fig.. 3d. We define

$$
U_t = \log(\text{HUR}_t)
$$
$$
T_t = \frac{\text{TAXREV}_t * \text{GDP}_t - \text{TAXREV}_{t-1} * \text{GDP}_{t-1}}{\text{TAXREV}_{t-1} * \text{GDP}_{t-1}}
$$
$$
G_t = \frac{\text{GDP}_t - \text{GDP}_{t-1}}{\text{GDP}_{t-1}}
$$
$$
I_t = \text{CPI}_t
$$

For this analysis we consider the annual data for 10 countries namely Australia, Canada, France, Germany, Italy, Japan, Korea, Mexico, Turkey, Great Britain and the United States of America for the period (2000 - 2019). We fit the following SCM:

$$T_t = f_T(t) + \epsilon_T$$
$$I_t = f_I(t) + \epsilon_I$$
$$G_t = f_G(T_t, I_t) + \epsilon_G$$
$$U_t = f_U(G_t, I_t) + \epsilon_U$$

by placing GPs on all functions $f_i(\cdot), i \in \{T, I, G, U\}$. This SCM is then used to generate interventional data and compute the values of $y_t^\star, t = 2010, \ldots, 2012$.

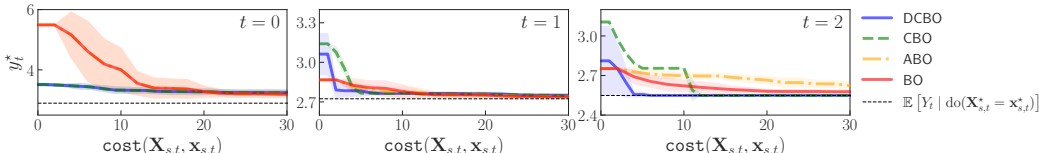

Figure 6: Experiment ECON. Convergence of DCBO and competing methods across replicates. The black line gives the optimal $y_t^*, \forall t$. Shaded areas are $\pm$ one standard deviation.

We run the optimization 10 times and plot the convergence path for DCBO and competing models (see Fig. 6). While all method perform similarly at $t = 2010$ and $t = 2011$, DCBO outperforms competing approaches at $t = 2012$. On average (see Table 1) DCBO finds the optimal intervention faster.

## 5.8   Results without convergence

We repeat all experiments in the paper allowing the algorithms to perform a lower number of trials at every time steps. This means that, for $t > 0$, when moving to step $t$ the convergence of the algorithm at step $t - 1$ is not guaranteed. In turn this affect the optimum value that the algorithm can reach at subsequent steps. Results are given in Table 1 and Table 2. The convergence paths for DCBO and competing methods are given in Fig. 7 to Fig. 11.

Table 1: Average modified gap measure (10 replicates) across time steps and for different experiments. See Fig. 1 for a summary of the compared methods. Higher values are better. The best result for each experiment is bolded. Standard errors in brackets.

|  | Synthetic data | | | | | | Real data | |
|---|---|---|---|---|---|---|---|---|
|  | STAT. | MISS. | NOISY | MULTIV. | IND. | NONSTAT. | ECON. | ODE |
| DCBO | **0.88** | **0.72** | **0.73** | **0.49** | 0.47 | **0.47** | 0.40 | **0.67** |
|  | (0.00) | (0.07) | (0.00) | (0.00) | (0.05) | (0.00) | (0.04) | (0.00) |
| CBO | 0.57 | 0.51 | 0.67 | 0.47 | 0.48 | **0.47** | **0.41** | 0.65 |
|  | (0.02) | (0.09) | (0.01) | (0.04) | (0.04) | (0.00) | (0.04) | (0.00) |
| ABO | 0.43 | 0.45 | 0.42 | 0.40 | **0.50** | 0.41 | 0.38 | 0.47 |
|  | (0.06) | (0.04) | (0.06) | (0.05) | (0.00) | (0.03) | (0.04) | (0.01) |
| BO | 0.42 | 0.41 | 0.41 | 0.38 | **0.50** | 0.40 | 0.40 | 0.46 |
|  | (0.06) | (0.05) | (0.07) | (0.07) | (0.01) | (0.04) | (0.04) | (0.03) |

## 5.9   Results over multiple datasets and replicates

In this section we show the results obtained running all the experiments in the main paper across 10 different observational dataset sampled from the SCM given above. Results are given in Table 3.

Table 2: Average percentage of replicates across time steps and for different experiments for which the optimal intervention set is identified. See Fig. 1 for a summary of the compared methods. Higher values are better. The best result for each experiment is bolded.

| | Synthetic data | | | | | | Real data | |
| | STAT. | MISS. | NOISY | MULTIV. | IND. | NONSTAT. | ECON. | ODE |
|---|---|---|---|---|---|---|---|---|
| DCBO | **90.0** | **70.00** | **93.00** | **93.33** | **96.67** | **66.67** | 73.33 | **33.33** |
| CBO | 76.67 | 63.33 | 76.67 | 86.67 | 93.33 | 33.33 | **80.00** | **33.33** |
| ABO | 0.00 | 0.00 | 0.00 | 0.00 | 100.00 | 0.00 | 66.67 | 0.00 |
| BO | 0.00 | 0.00 | 0.00 | 0.00 | 100.00 | 0.00 | 66.67 | 0.00 |

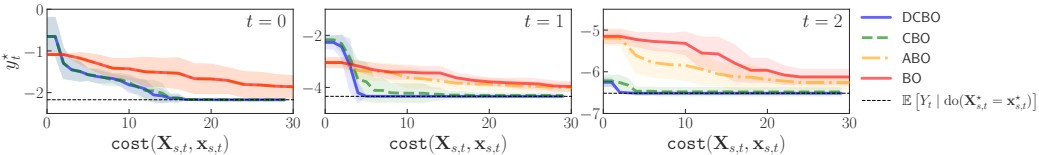

Figure 7: Experiment STAT. with maximum number of trials $H = 30$. Convergence of DCBO and competing methods across replicates. The black line gives the optimal $y_t^*$, $\forall t$. Shaded areas are $\pm$ one standard deviation.

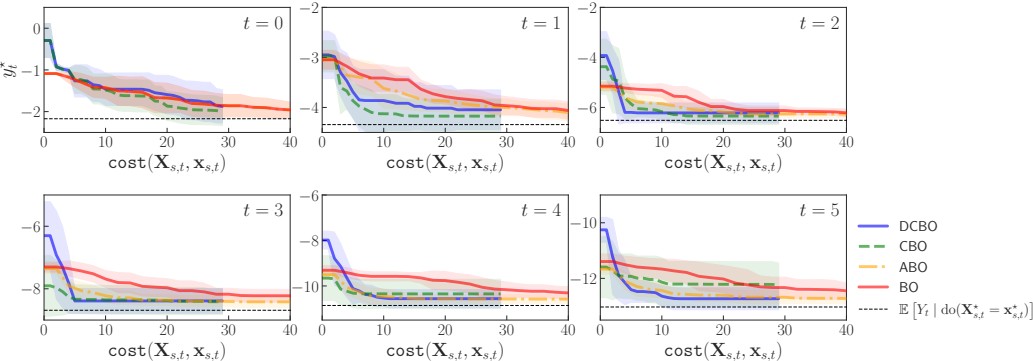

Figure 8: Experiment MISS. with maximum number of trials $H = 30$. Convergence of DCBO and competing methods across replicates. The black line gives the optimal $y_t^*$, $\forall t$. Shaded areas are $\pm$ one standard deviation.

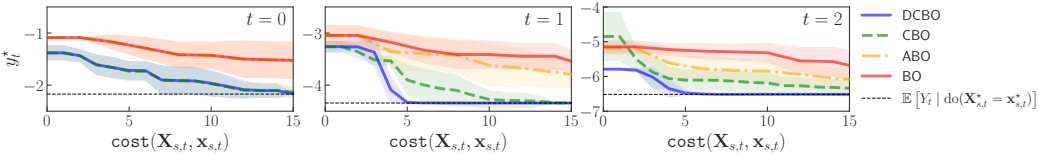

Figure 9: Experiment NOISY. with maximum number of trials $H = 30$. Convergence of DCBO and competing methods across replicates. The black line gives the optimal $y_t^*$, $\forall t$. Shaded areas are $\pm$ one standard deviation.

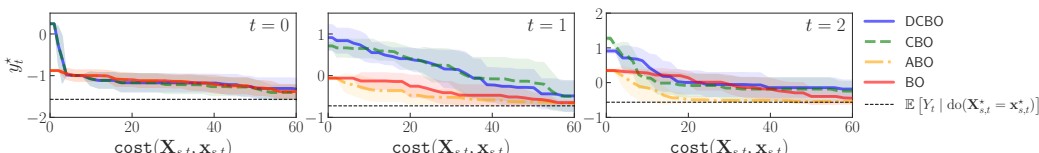

Figure 10: Experiment IND. with maximum number of trials $H = 30$. Convergence of DCBO and competing methods across replicates. The black line gives the optimal $y_t^*$, $\forall t$. Shaded areas are $\pm$ one standard deviation.

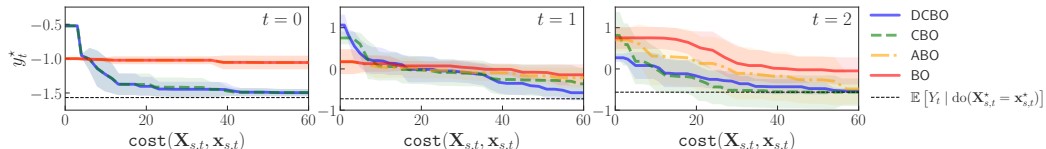

Figure 11: Experiment MULTIV. with maximum number of trials $H = 30$. Convergence of DCBO and competing methods across replicates. The black line gives the optimal $y_t^*$, $\forall t$. Shaded areas are $\pm$ one standard deviation.

Table 3: Average modified gap measure **across 10 observational datasets and 10 replicates**. Results are average figures across time steps. See Fig. 1 for a summary of the compared methods. Higher values are better. The best result for each experiment is bolded. Standard errors in brackets.

|  | \multicolumn{6}{c}{Synthetic data} |
|  | STAT. | MISS. | NOISY | MULTIV. | IND. | NONSTAT. |
|---|---|---|---|---|---|---|
| DCBO | **0.83** | **0.82** | **0.82** | **0.48** | 0.46 | 0.63 |
|  | (0.06) | (0.05) | (0.05) | (0.02) | (0.03) | (0.06) |
| CBO | 0.80 | 0.68 | 0.74 | **0.48** | 0.47 | **0.64** |
|  | (0.05) | (0.04) | (0.09) | (0.01) | (0.02) | (0.04) |
| ABO | 0.47 | 0.49 | 0.47 | 0.45 | 0.48 | 0.38 |
|  | (0.01) | (0.00) | (0.01) | (0.08) | (0.00) | (0.01) |
| BO | 0.47 | 0.47 | 0.47 | 0.40 | **0.50** | 0.38 |
|  | (0.01) | (0.01) | (0.01) | (0.07) | (0.00) | (0.01) |

# 6 Intervening on differential equations (ODE)

In this section we describe in detail the experiment conducted in §4.2. This example is based on the work by Blasius et al. (2020). In this demonstration we continue along that paradigm when we investigate a biological systems in which two species interact, one as a predator and the other as prey. Blasius et al. (2020) performed microcosm experiments (in a chemostat or bioreactor) with a planktonic predator–prey system.

We use the provided ODE (§6.2) from the paper (Blasius et al., 2020, Methods), which describes a stage-structured predator–prey community in a chemostat, as our SCM. As $\mathcal{D}^O$ we use the experimental data collected in vitro (for raw data see supplementary material of (Blasius et al., 2020)). The corresponding DAG (§6.3) and SCM (§6.4) is constructed from the ODE (see overleaf), by rolling out the temporal variable dependencies in the ODE (the idea is well illustrated in (Weber, 2016, Fig. 1)).

Using this setup we investigate a requisite intervention policy necessary to reduce the concentration of dead animals in the chemostat – $D_t$ in Fig. 3e.

## 6.1 Interpreting differential equations as causal models

A lot of work (Peters et al., 2020; Kaiser, 2016; Mooij et al., 2013; Bongers & Mooij, 2018; Hansen et al., 2014; Weber, 2016) has been dedicated to interpreting ordinary differential equations as structural causal models and consequently the associated task of intervening therein. More precisely, attention has been placed on extending causal theory (Pearl, 2000; Spirtes, 1995) to the cyclic case, thereby enabling causal modelling of systems that involve feedback (Mooij et al., 2013; Koster et al., 1996; Dechter, 1996; Neal, 2000; Hyttinen et al., 2012; Rubenstein et al., 2016; Peters et al., 2020).

Naively, the simplest extension to the cyclical case is by simply dropping the acyclicity constraint from the SCM (Mooij et al., 2013, §1). But then we are faced with a new problem: how do we "interpret cyclic structural equations" (Mooij et al., 2013)? The most common approach is to "assume an underlying discrete-time dynamical system, in which the structural equations are used as fixed point equations" (Mooij et al., 2013). This renders a simple schema wherein which we use the SCM as a set of updates rules, to find the values of the variables at $t + 1$, using the information from $t$. This is a popular paradigm, advanced by e.g. Spirtes (1995); Hyttinen et al. (2012); Dash (2005); Lacerda et al. (2012); Mooij et al. (2011). This is also the one we will use herein.

Another philosophy that deals with interventions in systems, was developed by Casini et al. (2011). In the same vein is the work by Gebharter (2014); Gebharter & Schurz (2016). This suite of work comes from the philosophy of science domain, rather than the statistical and machine learning literature, briefly reviewed in the previous two paragraphs. Theirs is primarily a concern with mechanisms (specifically "mechanistic biological models with complex dynamics" in the case of Kaiser (2016)) – fundamentally they are the same thing as our causal effects but the perspective is different. Casini et al. (2011) suggests that modelling (acyclical) mechanisms should be done by way of recursive Bayesian networks (RBN). Gebharter (2014) points out some shortcomings with Caisini's approach and proposes the multilevel causal model (MLCM) as a remedy. Notably though, both works assume acyclicity (and so cannot feature mechanisms with feedback) of the problem domain a shortcoming that Gebharter & Schurz (2016) deals with by extending the MLCM to allow for cycles. For completeness we should also say that the RBN was extended to handle cycles by Clarke et al. (2014) (their approach was used Gebharter & Schurz (2016) for extending the MLCM).

## 6.2 Ordinary differential equation

Blasius et al. (2020) develop a mathematical model, the set of ordinary differential equations in Eq. (17)–Eq. (22), to describe a stage-structured predator–prey community in a chemostat, which closely follows their experimental setup.

$$\frac{dN}{dt} = \delta N_{\text{in}} - F_P(N)P - \delta N \tag{17}$$

$$\frac{dP}{dt} = F_P(N)P - \frac{F_B(P)B}{\varepsilon} - \delta P \tag{18}$$

$$\frac{dE}{dt} = R_E - R_J - \delta E \tag{19}$$

$$\frac{dJ}{dt} = R_J - R_A - (m + \delta)J \tag{20}$$

$$\frac{dA}{dt} = \beta R_A - (m + \delta)A \tag{21}$$

$$\frac{dD}{dt} = m(J + A) - \delta D \tag{22}$$

A full description of all variables and parameters can be found in Table 4.

Table 4: Table describing variable and parameters of ODE in Eq. (17) – Eq. (22).

| Variable | Description | Value | Unit |
|---|---|---|---|
| $N$ | Nitrogen (prey) concentration | $\in \mathbb{R}^1$ | $\mu\text{mol} \cdot \text{N} \cdot \text{L}^{-1}$ |
| $P$ | Phytoplankton (predator) concentration | $\in \mathbb{R}^1$ | $\mu\text{mol} \cdot \text{N} \cdot \text{L}^{-1}$ |
| $E$ | Predator egg concentration | $\in \mathbb{R}^1$ | $\mu\text{mol} \cdot \text{N} \cdot \text{L}^{-1}$ |
| $J$ | Predator juvenile concentration | $\in \mathbb{R}^1$ | $\mu\text{mol} \cdot \text{N} \cdot \text{L}^{-1}$ |
| $A$ | Predator adult concentration | $\in \mathbb{R}^1$ | $\mu\text{mol} \cdot \text{N} \cdot \text{L}^{-1}$ |
| $D$ | Dead animal concentration | $\in \mathbb{R}^1$ | $\mu\text{mol} \cdot \text{N} \cdot \text{L}^{-1}$ |

| Parameter | Description | Value | Unit |
|---|---|---|---|
| $N_{\text{in}}$ | Nitrogen concentration in the external medium | 80 | $\mu\text{mol} \cdot \text{N} \cdot \text{L}^{-1}$ |
| $F_P$ | Algal nutrient uptake | - | $\mu\text{mol} \cdot \text{N} \cdot \text{L}^{-1}$ |
| $F_B$ | Rotifer nutrient uptake | - | $\mu\text{mol} \cdot \text{N} \cdot \text{L}^{-1}$ |
| $\varepsilon$ | Predator assimilation efficiency | 0.55 | - |
| $R_E$ | Egg recruitment rate | - | - |
| $R_J$ | Juvenile recruitment rate | - | - |
| $R_A$ | Adult recruitment rate | - | - |
| $m$ | Rotifer (predator) mortality rate | 0.15 | Per day |
| $\beta$ | Adult/juvenile mass ratio | 5 | - |

For additional details see (Blasius et al., 2020, Methods).

## 6.3 Corresponding directed acyclical graph

The original rolled-out DAG (Fig. 12b) is modified to remove graph cycles (Fig. 12c), where the corresponding dependencies are replicated in the SCM. Now, note first that the temporal roll-out of Fig. 12a contains no cycles (once the self-cycles have been re-purposed as temporal transition functions). Nonetheless, comparing Fig. 12b and Fig. 12c it can be seen that two edges have been removed to simplify the causal dependencies on the phytoplankton (predator) concentration i.e. to make it only dependent on the nitrogen concentration in the external medium as well as the most immediate predator concentration at time $t - 1$.

One large deviation from the original set of ODEs, is that we treat the $N_{\text{in}}$ as an instrument variable and moreover allow it to be manipulative. This means that in order to reduce the concentration $D_t$ we allow the optimisation frameworks to intervene also on $N_{\text{in}}$.

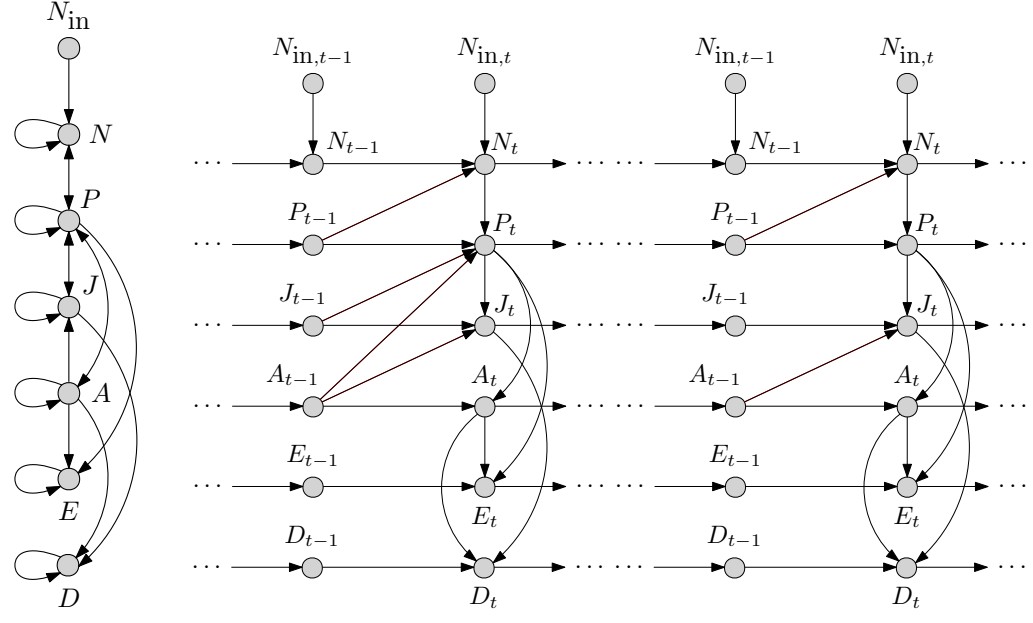

(a) ODE variable dependencies.

(b) First DAG approximation.

(c) Second DAG approximation.

Figure 12: Proposed time-indexed DAG representing the causal dependencies in the stage-structured predator–prey community in a chemostat. The vertices of the graph represent the concentrations of the different chemostat compounds at different discrete time points, where time is moving from left to right. Fig. 12a shows the variable dependencies as described in the original ODE found in Eq. (17) – Eq. (22) – notice the presence of self-loops and cycles. Figure 12b shows a first approximation to a corresponding causal graph, where the ODE has been 'rolled' out in time – note the absence of self-loops and cycles. Figure 12c shows a second approximation to the original ODE dynamics but this time removing two parent dependencies from $P_t$.

## 6.4 ODE as SEM

We fit the following SCM, based on the DAG in Fig. 12c:

$$N_{\text{in},t} = \epsilon_{N_{\text{in}}} \tag{23}$$
$$N_t = f_N(N_{\text{in},t}, N_{t-1}, P_{t-1}) + \epsilon_N \tag{24}$$
$$P_t = f_P(N_t, P_{t-1}) + \epsilon_P \tag{25}$$
$$J_t = f_J(P_t, J_{t-1}, A_{t-1}) + \epsilon_J \tag{26}$$
$$A_t = f_A(P_t, A_{t-1}) + \epsilon_A \tag{27}$$
$$E_t = f_E(P_t, A_t, E_{t-1}) + \epsilon_E \tag{28}$$
$$D_t = f_D(J_t, A_t, D_{t-1}) + \epsilon_D \tag{29}$$

by placing GPs on all functions $\{f_i(\cdot) \mid i \in \{N_{\text{in}}, N, P, E, J, A, D\}\}$. This SCM is then used to generate interventional data and compute the values of $\{d_t^\star \mid t = 0, 1, 2\}$.

Further, $\{\epsilon_j \sim \mathcal{N}(0,1) \mid j \in \{N_{\text{in}}, N, P, E, J, A, D\}\}$. We set $T = 3$, $N = 4$ where the manipulative variables are: $N_{\text{in},t}, J_t$ and $A_t$. This means in practise that we are interested in the start of the simulation where we are trying to reduce the mortality concentration, in the chemostat, from beginning where our observational samples $\mathcal{D}^O$ are formed from four time-series[2].

---

[2]We use data-files `C1.csv`, `C2.csv`, `C3.csv`, `C4.csv` from the original publication (Blasius et al., 2020) – available here: `https://figshare.com/articles/dataset/Time_series_of_long-term_experimental_predator-prey_cycles/10045976/1` [Accessed: 01/04/21].

Intervention domains are given by

$$D(N_{\text{in},t}) = [40.0, 160.0]$$
$$D(J_t) = [0.0, 20.0]$$
$$D(A_t) = [0.0, 100.0]$$

Notice that DCBO and CBO explore the set

$$\mathbb{M}_t = \{\{N_{\text{in},t}\}, \{J_t\}, \{A_t\}, \{N_{\text{in},t}, J_t\}, \{N_{\text{in},t}, A_t\}, \{J_t, A_t\}, \{N_{\text{in},t}, J_t, A_t\}\}$$

while BO and ABO will only intervene on $\{N_{\text{in},t}, J_t, A_t\}$. The optimal sequence of interventions is given by $\{\{J_0, A_0\}, \{M_1\}, \{M_2\}\}$.

Results are shown in Fig. 13. Note that the performance of DCBO and CBO are almost identical.

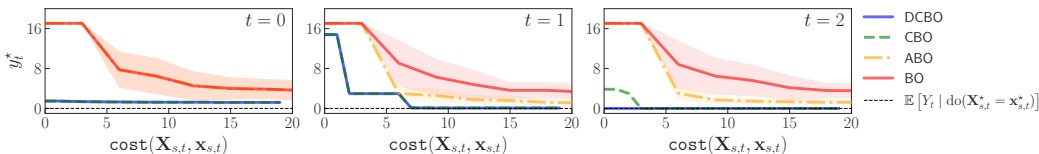

Figure 13: Experiment ODE with maximum number of trials $H = 20$. Convergence of DCBO and competing methods across replicates. The black line gives the optimal $y_t^*, \forall t$. Shaded areas are $\pm$ one standard deviation.

# 7 Applicability of DCBO to real-world problems

As previously done in CBO (Aglietti et al., 2020) and other causal decision-making frameworks (e.g. Bareinboim et al. (2015)) for static settings, in DCBO we assume to be able to repeatedly intervene in the system with interventions that have an instantaneous effect observed within the time slice duration. In other words, within every time step, we perform an intervention that changes the system and that leads to an effect for which we collect the corresponding target experimental value. However, the system reverts back once the experiment has been implemented and the agent can then explore alternative interventions and measure their effect too. In DCBO, the dynamics of the time resolution specified by the graph time indices is slower than the time you can take actions and see the effects.

While this assumption can be difficult to verify when interacting directly with the psychical world, it does not limit the applicability of the proposed framework to real-world problems. Indeed, in a variety of real-world settings, simulators or digital twins of real-world assets/processes are used in industrial settings and are fundamental in selecting actions before intervening in the real physical world. Digital twins provide virtual replicas of a physical object or system, such as a bridge or an engine, that engineers use for simulations before something is created or to monitor its operation in real-time. Examples are given by the digital twin of a 3D-printed stainless steel bridge (bri), NASA and U.S. Air Force vehicles (Glaessgen & Stargel, 2012), jet-engine monitoring, infrastructure inspection as well as cardiac medicine (vir). In all these settings, observational data are used to build the emulator which is "a living computer model which is continuously learning to imitate the physical world" (bri). We can then intervene on the digital twin to collect interventional data and measure the causal effects. Intervening in a simulator has a cost e.g. a computationally cost thus interventions need to be carefully picked by employing a probabilistic model that correctly quantifies uncertainty and integrates different sources of information. In DCBO this is done by using the dynamic causal GP model. Once an intervention has been implemented, the digital twin "reverts" to its unperturbed/observational nature (i.e. without intervention), allowing the user to investigate other interventions without having changed the "underlying state of the system" nor, indeed, the true system. Once an optimal intervention is found, the agent can implement it in the real system thus changing it. Note that our approach allows for noise in the likelihood function thus the simulator can be a noisy version of the physical world.

# 8 Connections

We conclude by providing a discussion of the links between DCBO, the two methodologies used as benchmarks in the experimental session, namely the CBO algorithm (Aglietti et al., 2020) and the ABO algorithm (Nyikosa et al., 2018), and the literature on bandits and RL. We discuss how their problem setups differ from our and highlight the reasons why DCBO is needed to solve the problem in Eq. (1).

**CBO algoritm** The CBO algorithm (Aglietti et al., 2020) can be used to find optimal interventions to perform in a causal graph so as to optimize a single target node $Y$. CBO addresses static settings where variables in $\mathcal{G}$ are i.i.d. across time steps, i.e. $p(\mathbf{V}_t) = p(\mathbf{V}), \forall t$, and only one static target variable exists. For instance, CBO can be used to find the optimal intervention for $Y$ in the DAG of Fig. 1b. In order to use CBO for the DAG of Fig. 1a, one would need to identify a unique target among $Y_{0:T}$, e.g. $Y_T$. However, optimizing $Y_T$ might lead to chose interventions that are sub-optimal for $Y_{0:T-1}$ thus not solving the problem in Eq. (1). In addition, to find the optimal intervention for $Y_T$, CBO explores all interventions in $\mathcal{P}(\mathbf{X}_{0:T})$ which results in a large search space and requires performing a high number of interventions. This slows down the convergence of the algorithm and increases the optimization cost. One can alternatively run CBO $T$ times optimizing $Y_t$ at each time step. Doing that would require re-initializing the surrogate models for the objective functions at every $t$ and would thus imply loosing all the information collected from previous interventions. Finally, in optimizing $Y_t$, CBO does not account for how the previously taken interventions have changed the system again slowing down the convergence of the algorithm. In order to recursively optimise intermediate outputs given the previously taken decisions one need to resort to DCBO. By changing the objective function at every time step, incorporating prior interventional information in the objective function and limiting the search space at every time step based on the topology of the $\mathcal{G}$, DCBO addresses the CBO issues mentioned above making it a framework that can be practically used for sequential decision making in a variety of applications.

**ABO algorithm**  While CBO tackles the causal dimension of the DCGO problem but not the temporal dimension, the ABO algorithm also addresses dynamic settings but does not account for the causal relationships among variables, see Fig. 1 for a graphical representation of the relationship between these methods. As in BO, ABO finds the optimal intervention values by breaking the causal dependencies between the inputs and intervening simultaneously on all of them thus setting $\mathbf{X}_{s,t} = \mathbf{X}_t$ for all $t$. Additionally, considering the inputs as fixed and not as random variables, ABO does not account for their temporal evolution. This is reflected in the DAG of Fig. 1(c) where both the horizontal links between the inputs and the edges amongst the input variables are missing. In solving the problem in Eq. (1) for the DAG in Fig. 1a, BO would disregard both the temporal dependencies in $Y$ and the input dependencies (DAG in Fig. 1d) while ABO would keep the former but ignore the latter. Differently from our approach, ABO considers a continuous time space and places a surrogate model on $Y_t = f(\mathbf{x}, t)$. $f(\mathbf{x}, t)$ is then modelled via a spatio-temporal GP with separable kernel. The ABO acquisition function for $f(\mathbf{x}, t)$ is then restricted to avoid collecting points in the past or too far ahead in the future where the GP predictions have high uncertainty. The spatio-temporal GP allows ABO to predict the objective function ahead in time and track the evolution of the optimum. However, in order for ABO to work the objective function rate of change over time must be slow enough to gather enough samples to learn the relationships in space and time. In our discrete time setting this condition is equivalent to ask that, at every time step, it is possible to perform different interventions with an underlying true function that does not change. Note that also in DCBO, Assumptions 1 imply a certain level of regularity in the objective functions. For instance, in the DAG of Fig. 1a, given that $\mathrm{Pa}(Y_t) = \{Z_t, Y_{t-1}\}, \forall t > 0$, the objective functions have a constant shape and are only shifted vertically by the performed interventions. While some regularity is also required in DCBO, through the causal graph we impose more structure on the objective function and its input thus lowering the need for exploration. The more accurate the estimation of the functions in the SCM is the more we can track the dynamic of the objective function and we can deal with sharp changes in the objectives. One additional important difference between ABO and DCBO is in the exploration of different intervention set. Indeed, by intervening on all variables, ABO can lead to sub-optimal solution. As mentioned for BO in Aglietti et al. (2020), depending on the structural relationships between variables, intervening on a subgroup might lead to a propagation of effects in the causal graph and a higher final target. In addition, intervening on all variables is cost-ineffective in cases when the same target can be obtained by setting only a subgroup of them. This is particularly true in the time setting as the optimal intervention set might not only be a subset of $\mathcal{P}(\mathbf{X}_t)$ but might also evolve overtime.

**Bandits and RL**  In the broader decision-making literature, causal relationships have been previously considered in the context of multi-armed bandit problems (MAB, Bareinboim et al., 2015; Lattimore et al., 2016; Lee & Bareinboim, 2018, 2019) and reinforcement learning (RL, Lu et al., 2018; Buesing et al., 2018; Foerster et al., 2018; Zhang & Bareinboim, 2019; Madumal et al., 2020). In these cases, the actions or arms correspond to interventions on an arbitrary causal graph where there exists complex links between the agent's decisions and the received rewards. Causal MAB algorithms focus on static settings where the distribution of the rewards is stationary and is not affected by the pulled arms. In addition, MAB focus on intervention on discrete variable and only deal with the problem of selecting the right intervention set but not the intervention value. Differently from DCBO, RL algorithms explicitly model the state dynamic and account for the way each action affect the state of the environment. DCBO setting differs from both causal RL and causal MAB. DCBO does not have a notion of *state* and therefore does not require an explicit model of its dynamic. The system is fully specified by the causal graph and the connected structural equation model. As in BO, DCBO does not aim at learning an optimal policy but rather a set of optimal actions. Furthermore, within each time step, DCBO allows the agent to perform a number of explorative interventions which are not modifying the environment. Once the optimal action is identified this is propagated in the system thus changing it. Differently from both MAB and RL, DCBO is *myopic* that is interventions are decided by maximizing the one-step ahead utility function. We leave the integration of DCBO with a non-myopic BO scheme to future work.