# OpenReview forum: "Dynamic Causal Bayesian Optimization"
_NeurIPS.cc/2021/Conference — NeurIPS 2021 Poster_

### Official Review · Reviewer_o1ra · 2021-07-16

**Rating:** 6
**Confidence:** 5

**Summary:**

In this paper, the authors extend the notion of casual Bayesian optimisation from a static to a dynamic scenario which, indeed, has many relevant applications. The paper theoretically analyses the proposed approach and run the algorithm in a set of real-world and synthetic data. This work is really interesting but I have some questions related to scalability and applicability that I hope the authors could help me understand.

**Limitations And Societal Impact:**

I think the paper can benefit from a larger study of social impacts especially after motivating health-care in the abstract.

**Main Review:**


1) I found definition 1 a bit confusing in some places:
1.1.) When the set of observed variables V are defined, the authors write that those variables in the model are determined by variables in U \cup V. So, the variables in V are defined by variables in the union of U and V. I am a bit confused about this as this somehow defies the set V through the set V.
1.2.) When the set of functions f_i is introduced, the authors wrote that each of those is mapping from the union U_i and Pa(U_i) to V_i, where Pa(U_i) is a subset of V. Hence, from this definition, it follows that mapping f_i for each element in the union U_i and Pa(U_i) should put in correspondence a value of variable V_i. However,  f_i(Pa(u_i), u_i) is then defined as a function for an ordered pair of elements Pa(u_i) and u_i. Wouldn't this mean that the function f_i is defined on the cartesian product Pa(U_i) x U_i rather than the union P(U_i) \cup U_i?
1.3.) When the authors define the entire set F of functions f_1,.... f_n it is said that F forms a mapping from U to V. But each f_i is mapping not just from U but also it takes as an input element from Pa(U)\in V. Hence, should not F be a mapping from U x V to V?

2) Similar to standard assumptions in causal BO, the authors assume a known graph G. It is already rather hard to satisfy this assumption is standard causal BO without explicit prior knowledge. In this paper, does not this assumption entail even more prior knowledge? Shouldn't we also know how the graphs are varying over time as well?

3) The algorithm proposed is interesting. I am wondering if the authors might reframe this as an instance of reinforcement learning and solve the corresponding problem? In other words, can there exist any relations to reinforcement learning and/or dynamic programming? If so, how can I think of the policy? Would the current algorithm correspond to a greedy policy for example?

4) In formulation (1) for DCGO, if we are allowed to execute interventions on all variables X_t, will it not be the best option for us to optimise Y_t? The reason I ask is that although the cost of intervention is introduced it is not a involved in the objective function (but involved in the acquisition function)?

4) Assumption 2, how restrictive is the additive assumption? In the experiments do we consider problems with such function f_{Y_t} and if yes, do we have a mechanism to check this property somehow?

5) How the authors restrict the search space was not clear to me. My confusion stems from assumption 1, wherein the dynamic case the search space of subsets is the same as in the static case. Still, it is not clear, what is the size of this collection in this static case? Is it the cardinality of the power set, i.e. 2^{|X_t|}?

6) Maybe removing the and operation from line 217 can aid readability.

7) Can the authors elaborate/study the effect of the topology of the underlying DAG graph on the overall performance of the algorithm?



**Time Spent Reviewing:**

4

---

> ### Author Response · Authors · 2021-08-10
> **Response to Reviewer o1ra**
>
> Thank you for your detailed and helpful feedback. We address each of your comments below, in point and order.
>
> - (1.1). Each variable in the set $\mathbf{V}$ of the SCM is determined by exogenous variables ($\mathbf{U}$) and *other* endogenous variables in the graph. For instance, in the DAG of the first row of Fig. 2, we have $\mathbf{V}_0 = \{X_0,Z_0,Y_0\}$ and the variable $Z_0$ is determined by $U_0$ and $X_0$ which also belongs to $\mathbf{V}_0$ for $t=0$. We provide a table summarizing the used notation in the appendix.
>
> - (1.2) and (1.3). Thank you for picking up this error, we have now corrected our typo. Instead of $Pa(U_i)$ we should have $Pa(V_i)$. The variables in $\mathbf{U}$ are exogenous and, by definition, do not have any parents in the causal graph.
>
> - (2). Specifying the causal graph requires some prior domain knowledge both in DCBO and CBO. One of the assumptions that we make in this paper is the invariance of the causal structure *within* every time-slice. The user only needs to specify the time transitions (horizontal directed edges in the graph) but, similarly to CBO, only one graph structure is kept constant across time slices. However, this does not limit the complexity of the causal effects we can capture and track efficiently, see response to R \# YSDW (4). In addition, notice that, while the structure remains the same, the causal effects can change as the functional relationships in the SCM can vary freely. As discussed in Section 2 and Section 5, integrating the DCBO framework with a causal discovery method or accounting for the unknown graph in the surrogate model formulation is an important research direction that we are actively investigating.
>
> - (3). Connecting DCBO and RL (and possibly causal RL) ultimately hinges to the known connection between BO and RL. In Section 7 of the supplementary material we provide a longer discussion of the differences between the proposed framework and bandits and RL. Notice that here, as in standard BO, we select actions based on the one-step ahead EI therefore we act greedily. Extending DCBO to use a non-myopic acquisition function such as the one proposed in [33] or [34] would allow the algorithm to select interventions in terms of multiple-step ahead reward and we will discuss this further in the paper. While this would make DCBO closer to RL, note that, as in BO, in DCBO we optimize a final reward. Linking DCBO to RL by writing the expected utility in the form of a Bellman equation (as done for non-myopic BO in [34]) is an important area of future work that would connect different approaches in the sequential causal decision making literature. %We have further commented on this aspect.
>
> - (4). This specific aspect has been previously investigated in CBO [2] where the authors show how intervening on all variables might lead to sub-optimal results. Indeed, this might be cost inefficient and might block the propagation of the causal effects therefore allowing to reach sub-optimal solutions. This was also shown in [19] for structural causal bandits. We have clarified this further in the paper. Regarding the cost inefficiency, notice that if the same function value can be achieved by a cheaper intervention on a subset of variables we would like the acquisition function to avoid the exploration of the set $\mathbf{X}_t$. In other words, we do not want to change the objective function but rather maximize the EI per unit of cost which is exactly what we compute by dividing with the cost.
>
> - (5). The additive structure of the functional relationship for $Y$ is needed to construct the prior distribution for the surrogate model exploiting the theoretical results. These indeed hold when Assumptions 1 are verified and enables the transfer of interventional data observed across different time steps. We have used this functional form for all the experiments analysed in the paper. In the causality literature, Gaussian processes models are well established and have shown good performances compared to parametric linear and non-linear models (see e.g. [35], [36] and [37]). Despite the additive assumption, the sum of GPs gives a flexible and computationally tractable model that can be used to capture highly non-linear causal effects while helping with interpretability, see [3] and [4]. We have further discussed this assumption in the paper. We do not know of any explicit method that allows one to check for the functional compositional nature of the true underlying generative model $f(Y_t)$. However, methods have been developed (see [5]), for GPs, wherein which an automatic kernel selection/discovery takes place as part of the learning task such that the best kernel composition prior is learned, prior to the main task. We have mentioned this additional direction as future work.
>
> - (6). In the static case, CBO [2] identifies sets worth intervening on by exploiting the results in [19]. These are the so called minimal intervention sets and are denoted by $\mathbb{M}$. Therefore, in the static setting, CBO explores $2^{|\mathbb{M}|}$ which is lower than $2^{|\mathbf{X}|}$. A similar approach can be used for DCBO. Instead of exploring $2^{|\mathbf{X}_t|}$ we can explore $2^{|\mathbb{M}_t|}$ where $\mathbb{M}_t$ can be naively derived for every time step following [19]. Proposition 3.1 further simplifies the identification of the sets worth intervening on by showing that $\mathbb{M}_t = \mathbb{M}_0$. One only needs to compute the minimal intervention sets once as these stay constant across time slices. We will clarify this point in the paper.
>
> - (7) Thank you for your suggestion. We will replace it.
>
> - (8). The experimental results shown in the paper provide an empirical analysis of the effect of a variety of different graph structures on the performance of the algorithm. Indeed, we have considered both simple graphs where a causative structure among the inputs exists (DAG in Fig. 1 (a)) but also where the inputs are independent (DAG in Fig. 3 (b)). In addition, we have considered DAGs with higher number of variables (DAGs in Fig. 3 (a), (d), (e)) and a DAG where the parents for some variables change over time thus making it non-stationary (DAG in Fig. 3 (c)). We have commented on the results across graphs showing, for instance, that when there is no causal relationship among inputs, considering a causal framework might slow down the convergence. The opposite is true when complex causal links existing among inputs and outputs. In the absence of unobserved confounders, the analysed DAG structures capture the most significant cases and those that might affect DCBO performance. A theoretical analysis of the impact of the graph structure on convergence is beyond the scope of this paper.
>
> We have added a comment on the societal impacts for health-care applications.
>
> [2] Aglietti, V., Lu, X., Paleyes, A. and González, J., 2020, June. Causal Bayesian Optimization. In International Conference on Artificial Intelligence and Statistics (pp. 3155-3164). PMLR.
>
> [3] Duvenaud, David. Automatic model construction with Gaussian processes. Diss. University of Cambridge, 2014.
>
> [4] Duvenaud, D., Nickisch, H., \& Rasmussen, C. E. (2011). Additive Gaussian processes. arXiv preprint arXiv:1112.4394.
>
> [5] Duvenaud, D., Lloyd, J., Grosse, R., Tenenbaum, J., \& Zoubin, G. (2013, May). Structure discovery in nonparametric regression through compositional kernel search. In International Conference on Machine Learning (pp. 1166-1174). PMLR.
>
> [19] Lee, Sanghack, and Elias Bareinboim. Structural causal bandits: where to intervene?. Advances in Neural Information Processing Systems 31 31 (2018).
>
> [33] González, J., Osborne, M., \& Lawrence, N. (2016, May). GLASSES: Relieving the myopia of Bayesian optimisation. In Artificial Intelligence and Statistics (pp. 790-799). PMLR.
>
> [34] Jiang, S., Chai, H., Gonzalez, J., \& Garnett, R. (2020, November). BINOCULARS for efficient, nonmyopic sequential experimental design. In International Conference on Machine Learning (pp. 4794-4803). PMLR.
>
> [35] Silva, R., \& Gramacy, R. B. (2010). Gaussian process structural equation models with latent variables. arXiv preprint arXiv:1002.4802.
>
> [36] Witty, S., Takatsu, K., Jensen, D., \& Mansinghka, V. (2020, November). Causal inference using Gaussian processes with structured latent confounders. In International Conference on Machine Learning (pp. 10313-10323). PMLR.
>
> [37] Zhang, K., Schölkopf, B., \& Janzing, D. (2012). Invariant gaussian process latent variable models and application in causal discovery. arXiv preprint arXiv:1203.3534.

---

### Official Review · Reviewer_Rw3m · 2021-07-18

**Rating:** 6
**Confidence:** 2

**Summary:**

This paper works with the Dynamic Causal Global Optimization problem. For a temporal structural causal model, which is a sequence of structural causal models with causality in time, meaning there are dependencies on the values of variables from previous step. The goal is to optimize the outcome variable by manipulating some observable variables (those are chosen by the model).

An algorithm called dynamic causal bayesian optimization is proposed to solve such a problem. The objective function $f_{s,t}$ is decomposed into the sum of a function depending on the optimized observed variables and another function depending on others (variables in the current time). Thus the optimization can be done step by step in time.

A dynamic causal GP model is introduced for practically implement the DCBO algorithm.

For experiments, both Synthetic and Real data are used, in several different structural causal models, to compare the behavior of DCBO, to that of CBO, ABO and BO.

**Limitations And Societal Impact:**

Yes

**Main Review:**

(I) The problem and methods are noval as far as I know.

(II) The results are well supported. The methods used are appropriate. It is a complete work.

(III) The paper is well organized, the structure is clear, important conclusions and concepts are provided in math environments (theorem / lemma / corollary / definition). All conclusions have formal proofs. However, there are some typos and font mismatches in the text and formulas.

(IV) This work solves the dynamic problem and builds a relation to previous CBO problem. It is difficult for me to determine how important this work is, because of my lack of experience in this area.

----------------------
Questions:
- The conclusion depends on the assumptions. So:
   - How likely the additivity of $f_{Y_t}$ is satisfied? What is the difficulty in solving the problem without additivity (I guess, the theorem 1 will no longer be of the same format, the sum of $f^Y$ and $f^{NY}$).
   - What will happen if the structure of causal model changes in time? (It is assumed $G(t)=G(0)$ in the assumption).
- The practical algorithm depends on Gaussian Process, how representative is GP in this problem?
- Is it possible to use other surrogate models, like deep GP?

**Time Spent Reviewing:**

18

---

> ### Author Response · Authors · 2021-08-10
> **Response to Reviewer Rw3m**
>
> Thank you for your detailed and helpful feedback. We address each of your comments below, in point and order.
>
> - (1). *Functional Assumption*. As discussed for R\# o1ra (5), the additive structure of the functional relationship for $Y$ is needed to derive the theoretical results and exploit them to construct the prior distribution for the surrogate model. In turns, this enables the transfer of interventional data observed across different time steps. When this is not the case, a standard causal GP prior (like the one used in CBO) or an acausal GP prior would have to be used thereby slowing down the convergence of the algorithm. Despite this assumption, the use of GPs allow us to have flexible models with highly non-linear causal effects across different graph structures.
>
> - (2). *Graph Assumption*. For clarity this discussion regards changing the causal structure **within** a time-slice. Alas, in such a case Theorem 1 would have to be extended to account for a changing causal graph. While this is possible, it is not straightforward as a varying graph would lead to the existence of different functions and search spaces across time steps. This might thus limit the possibility to transfer previously found optimal interventions. A formulation of Theorem 1 for more general graph structures is an open problem that we leave for future research.
>
> - (3). *GP and alternative models*. As in standard BO, other surrogate models such as e.g. random forests [38], student t-processes [39] and neural networks [40] could be used. Among these, Deep GP could be used for DCBO and have indeed been used for standard BO, see [41]. However, alternative surrogate models would pose additional computational challenges and might require resorting to approximate inference schemes slowing down the algorithm. In all analysed settings we found GPs to provide enough flexibility and to correctly quantify uncertainty while allowing for closed form updates of the posterior distribution.
>
> [38] Hutter, F., Hoos, H. H., \& Leyton-Brown, K. (2011, January). Sequential model-based optimization for general algorithm configuration. In International conference on learning and intelligent optimization (pp. 507-523). Springer, Berlin, Heidelberg.
>
> [39] Shah, A., Wilson, A., \& Ghahramani, Z. (2014, April). Student-t processes as alternatives to Gaussian processes. In Artificial intelligence and statistics (pp. 877-885). PMLR.
>
> [40] Snoek, J., Swersky, K., Zemel, R., \& Adams, R. (2014, June). Input warping for bayesian optimization of non-stationary functions. In International Conference on Machine Learning (pp. 1674-1682). PMLR.
>
> [41] Hebbal, A., Brevault, L., Balesdent, M., Talbi, E. G., \& Melab, N. (2019). Bayesian optimization using deep Gaussian processes. arXiv preprint arXiv:1905.03350.

---

### Official Review · Reviewer_YSDW · 2021-07-18

**Rating:** 7
**Confidence:** 3

**Summary:**

The paper proposes a model, Dynamic Causal Bayesian Optimization (DCBO), which builds upon existing models to find a sequence of interventions , optimizing the target variable at each time step in a causal system. It combines Causal Bayesian Optimization, which builds around casual information and finds the optimal intervention in a DAG without temporal evolution, with Dynamic Bayesian Network, which is used in time-series modelling and carries dependence assumptions that don’t imply causation. DCBO represents associations between features and the causality between input and outputs, both of which can change over time. The modeling therefore offers a novel approach for decision making in dynamic systems ie. find optimal intervention at time t. The paper compares the model’s results against algorithms which either account for temporal evolution or optimize the sequence of interventions of a system (as no model exists which does both). The algorithm is run on both interventional and observational data. Finally, synthetic experiments are performed, such as creating noisy variables, incorporating missing observational data, and creating non-stationary settings.

**Limitations And Societal Impact:**

One of the paper's main points is to show the advantage of introducing a dynamic extension to existing causal Bayesian optimization methods. However, in line 145, invariance of the causal structure is assumed. This is a really restrictive hypothesis that severely hinders the possible application of this framework.
To be fair, the submission states that some of the issues we point out will be addressed in future work. However, this framework could not be applied to real-world problems without such work.

**Main Review:**

The paper describes the modeling with mathematical rigor and precision. However, given the assumptions with which the framework is built, it is doubtful whether the approach could be used in any real-world scenario. Specifically, the modeling requires that the optimizing agent is allowed to perform a number of explorative interventions which do not change the underlying state of the system. It is difficult to identify a use case that satisfies this requirement. In section 4.2 (Line 342-351), the author provides “real experiments” that empoy real data to train a simulator/a system of ODE, which is then optimized. This is not an example of a “real experiment”.
The submission also does not elucidate how the proposed model identifies optimal interventions or treatment over a given time horizon.
The theoretical results obtained for both the synthetics and the real dataset do not show a substantial improvement from the already existing casual optimization models, with only slight improvements.
On a minor point, the mathematical notations are a bit cumbersome, and it is easy to get lost in a great variety of subscripts and quotes. Including a table summarizing the notation in one place would be particularly helpful.

**Time Spent Reviewing:**

3

---

> ### Author Response · Authors · 2021-08-10
> **Response to Reviewer YSDW**
>
> Thank you for your detailed and helpful feedback. We address each of your comments below, in point and order.
>
>
> - (1). *Applicability to real-world problems*. Thanks for bringing up this point which is central to DCBO. We have now added additional material discussing this point in greater detail.
>   - The sentence in line 46 is indeed imprecise and might lead to misunderstanding. As specified later in the paper, we do not have a notion of state in DCBO (see line 86 of the paper) and instead of  "underlying state" we should have said "underlying function" or "underlying causal effect". We have corrected it in the paper.
>   - As in CBO, in DCBO we assume that, within a time slice, we can repeatedly intervene in the system with interventions that have an instantaneous effect observed within the time slice duration. In other words, within every time step, we perform an intervention that changes the system and that leads to an effect for which we collect the corresponding target experimental value. However, the system reverts back once the experiment has been implemented and the agent can then explore alternative interventions and measure their effect too. The dynamics of the time resolution specified by the graph time indices is slower than the time you can take actions and see the effects. This assumption has been previously used in CBO [2] and in other causal decision-making frameworks such as for structural causal bandits [19].
>   - In terms of *real word applications*, we would like to point to the variety of real-world settings where simulators are used. Indeed, emulators or digital twins of real-world assets/processes are used in industrial settings as we speak and are fundamental in selecting actions before intervening in the real physical world. Digital twins provide virtual replicas of a physical object or system, such as a bridge or an engine, that engineers use for simulations before something is created or to monitor its operation in real-time. Examples are given by the digital twin of a 3D-printed stainless steel bridge [3], NASA and U.S. Air Force vehicles [4], jet-engine monitoring, infrastructure inspection as well as cardiac medicine [5]. In all these settings, observational data are used to build the emulator which is ''a living computer model which is continuously learning to imitate the physical world'' [3]. We can then intervene on the digital twin to collect interventional data and measure the causal effects.
>   - Intervening in a simulator has a cost e.g. a computationally cost thus interventions need to be carefully picked by employing a probabilistic model that correctly quantifies uncertainty and integrates different sources of information. In DCBO this is done by using the dynamic causal GP model. Once an intervention has been implemented, the digital twin `reverts' to its unperturbed/observational nature (i.e. without intervention), allowing the user to investigate other interventions without having changed the ``underlying state of the system'' nor, indeed, the true system. Once an optimal intervention is found, the agent can implement it in the real system thus changing it. Note that our approach allows for noise in the likelihood function thus the simulator can be a noisy version of the physical world. We have discussed these points further in the paper, clarifying the use of simulators and commenting on the use of DCBO for real-world settings.
>   - Finally, note that the focus of this paper is not the construction of a reliable emulator for a real-world system. We demonstrate and showcases how one can take decisions once the simulator has been created resorting to domain knowledge. Future collaborations with practitioners across different fields are crucial for further developing DCBO.
>
> - (2). *Selection of optimal interventions*. Interventions are selected by resorting to CBO and using the proposed surrogate model. This is discussed in lines 241-254, where we also mention the used acquisition function specified in Algorithm 1. We have clarified this point in Section 3.3 and Algorithm 1.
>
> - (3). *Results*. We believe the reviewer is referring to the experimental results as the theoretical results are not data or problem-dependent. Notice that, in terms of GAP metric, DCBO outperforms CBO in all settings. Similar results are observed for the experiments included in the appendix. Aside from the numerical results, the theoretical contributions and the developed modelling approach are a significant step towards a sequential causal decision-making algorithm that can be used in practice and in a variety of settings.
>
> - (4). *Assumptions*. We assume invariance of the causal structure only *within* each time slice. This means that, while the graph is constant among the variables for every $t$, across time steps both the graph and the functional relationship can change. Therefore, not only can the causal effects change significantly across time steps, but also the input dimensionality of the causal functions we model. For instance, in the DAG of Fig. 3 (c), the target function for $Y_2$ has dimensionality 3 and a function $f_{Y_t}(\cdot)$ that is completely different from the one assumed in step $t=1$ where $Y_1$ has only two parents. We can thus model a wide variety of settings despite this assumption as confirmed by the various SCMs considered in the experiments. We have further commented on this in the paper.
>
> We have extended the nomenclature table provided in Section 1 of the appendix (see page 2) so as to further clarify notation and aid the reader.
>
> [2] Aglietti, V., Lu, X., Paleyes, A. and González, J., 2020, June. Causal Bayesian Optimization. In International Conference on Artificial Intelligence and Statistics (pp. 3155-3164). PMLR.
>
> [19] Lee, S. and Bareinboim, E., 2018. Structural causal bandits: where to intervene?. Advances in Neural Information Processing Systems 31, 31.
>
> [3] Digital twin of the world’s first 3D printed stainless steel bridge. Retrieved: August 7th, 2021. URL: https://www.turing.ac.uk/research/research-projects/digital-twin-worlds-first-3d-printed-stainless-steel-bridge.
>
> [4] Glaessgen, E., \& Stargel, D. (2012, April). The digital twin paradigm for future NASA and US Air Force vehicles. In 53rd AIAA/ASME/ASCE/AHS/ASC structures, structural dynamics and materials conference 20th AIAA/ASME/AHS adaptive structures conference 14th AIAA (p. 1818).
>
> [5] Creating a virtual replica. Retrieved: August 7th, 2021. URL: https://www.ingenia.org.uk/getattachment/dc398efc-4995-46b8-ad8b-8414a413b27a/Digital-twins.pdf.

---

> > ### Comment · Reviewer_YSDW · 2021-09-01
> > **Thank you for the diligent response and apologies for the delay**
> >
> > After reading the other reviews and your responses, I have raised my score. I commend the authors for their hard work.

---

### Official Review · Reviewer_Ct7A · 2021-07-28

**Rating:** 7
**Confidence:** 3

**Summary:**

This paper addresses the Dynamic Causal Global Optimization (DCGO) problem that aims to find both the optimal set of manipulative variables and the optimal intervention levels to the selected variables. This is a time-series optimization that determines the best way to intervene in the present based on the results of interventions made in the past, and is considered to be a formulation of a particularly important problem e.g. in the medical field.
The authors make three assumptions about the DCGO problem:
(1) the causal structure is invariant with respect to time,
(2) the function value of a variable Y_t is determined from the function value of its parent node, and the function value of the parent node can be expressed as the sum of the function value of the target node and that of the non-target node,
(3) the DAG does not contain any unobserved confounding factors.

Under these assumptions, it follows that the objective function of DCGO at time t can be expressed recursively using the results of interventions up to time t-1, and that the search space of DCGO is invariant with respect to time.
Under the above problem setup, the proposed method DCBO is an EI-based Bayesian optimization algorithm based on the Dynamic Causal Gaussian Process model whose mean and covariance function are defined by the recursive representation of the objective function.

Experiments to validate the effectiveness of the proposed method have been conducted on both synthetic and real data. Experiments with synthetic data have been conducted in six situations: when DAGs and SCMs are stationary/non-stationary with respect to time, when the manipulative variables contain noise, when there are missing observations, when there is no causal relationship among manipulative variables,
and when there are multiple intervening variables.
Two real data experiments were conducted: one on the problem of minimizing the unemployment rate in a closed economy (economics), and the other on the problem of finding an intervention that would reduce the phenomenon of a concentration of dead animals in a chemostat (biology).

**Limitations And Societal Impact:**

The authors also consider experiments in problem settings where the proposed method does not work well.
I believe that there is no negative social impact from this research.

**Main Review:**

Originality: This paper aims at the optimization of sequential interventions, i.e., identifying the variables to be intervened and determining the optimal way to intervene.
This involves important problems, such as stage-by-stage optimization of multi-stage treatments in medicine. I think it is an important work to formulate this as an optimization problem under uncertainty, and to propose a framework for automatically searching for optimal interventions using a Bayesian optimization-type algorithm.

Quality: This paper technically sounds and the performance of the proposed method has been verified in various situations including real data. In particular, settings where there is no causal relationship between the manipurative variables that would disadvantage the proposed method are also considered (in fact, the existing methods outperforms the proposed method in terms of accuracy), and thus it seems that fair comparisons are conducted. Theoretically, two properties Theorem 1, which states that the objective function can be written recursively, and Proposition 3.1, which states that the search space is time-invariant, may be important in Bayesian optimization, a problem that involves optimizing a function that is expensive to begin with.

Clarity: This paper is well organized. It clearly explains problem formulation, why a general framework for dynamic causal global optimization is needed, the process of deriving the proposed method, and its theoretical considerations.

Significance: As mentioned above, the problem considered in this study may have important applications. For example, in terms of medical applications, a reinforcement learning-based method called "dynamic treatment regime" is being considered as another approach to sequential optimization of interventions. However, this method has several problems that need to be solved before it can be put to practical use, such as being complicated of the method itself and the difficulty of estimation. There are still very few examples of its application to real problems.
It is expected that this and subsequent studies will provide a practical approach to these problems.

Questions

- How realistic are the three assumptions (especially the first one that the causal structure is invariant with respect to time) introduced in section 2? Also, is it correct to interpret that the non-stationary case considered in the experiment deals with a situation that violates this assumption? In that case, I think the theoretical properties such as recursive representation of the objective function and constant search space are not guaranteed. How can we treat this?

- How do you determine the initial state \mathbb{M}_0　of the search space?

- In the equation that define the GP prior, the notation "m_{s, t}=... \wedge k_{s, t}=..." is confusing and better to be changed.
e.g. m_{s, t}=..., k_{s, t}=...

**Time Spent Reviewing:**

20h

---

> ### Author Response · Authors · 2021-08-10
> **Response to Reviewer Ct7A**
>
> Thank you for your detailed and helpful feedback. We address each of your comments below, in point and order.
>
> - (1). *Assumptions*. As mentioned for R\# YSDW (4), we assume invariance of the causal structure only *within* each time-slice (i.e. the structure, edges and vertices, concerning the nodes with the same index $t$ of the CGM). This means that, while the graph is constant among the variables, across time steps both the graph and the functional relationship can change. Therefore, not only does the causal effects change significantly across time steps, but also the input dimensionality of the causal function we model change. For instance, in the DAG of Fig. 3 (c), the target function for $Y_2$ has dimensionality 3 and a function $f_{Y_t}(\cdot)$ that is completely different from the one assumed for $Y_1$ that has only two parents. We can thus model a wide variety of settings and causal effects despite this assumption. Specifically, in the non stationary experiments this assumption is still verified. When this is not the case, Theorem 1 does not hold and one would need to revert back to a standard GP prior or a causal GP prior as the one used by CBO [2].
>
> - (2). *Search space*. In the static case, CBO [2] identifies sets worth intervening on by exploiting the results in [19]. These are the so called minimal intervention sets (MISs) and are denoted by $\mathbb{M}_0$ for $t=0$. They are identified by finding sets such that their causal effect cannot be expressed as the causal effects of a subset of intervention variables. See Section 3.1 in CBO [2] for more details. We have clarified this point in Section 3.2 of the paper.
>
> - (3) We have changed the wedge notation as suggested.
>
> [2] Aglietti, V., Lu, X., Paleyes, A. and González, J., 2020, June. Causal Bayesian Optimization. In International Conference on Artificial Intelligence and Statistics (pp. 3155-3164). PMLR.
>
> [19] Lee, S. and Bareinboim, E., 2018. Structural causal bandits: where to intervene?. Advances in Neural Information Processing Systems 31, 31.

---

> > ### Comment · Reviewer_Ct7A · 2021-09-02
> > **thanks to the authors' response**
> >
> > Thank you for responding to the review comments.
> > I understand that the authors have given me clear answers to two of my main concerns (about assumptions and initialization).
> > Although I will not change my score, I recommend again to accept.

---

### Decision · Program_Chairs · 2021-09-27

**Decision:**

Accept (Poster)

**Comment:**

This paper addresses an extension of causal Bayesian optimization from static to dynamic scenario to find a sequence of optimal interventions in a dynamic causal system. The problem formulation and the method are novel. In addition,  the results are well supported by experiments.  In general, the paper is well written. All of reviewers agree that the paper contains interesting contributions that are deserved to be presented in the conference.